# Multi-scale molecular dynamics simulations of enhanced energy transfer in organic molecules under strong coupling

Ilia Sokolovskii[1,3], Ruth H. Tichauer [1,2,3], Dmitry Morozov [1], Johannes Feist [2] & Gerrit Groenhof [1]✉

Exciton transport can be enhanced in the strong coupling regime where excitons hybridize with confined light modes to form polaritons. Because polaritons have group velocity, their propagation should be ballistic and long-ranged. However, experiments indicate that organic polaritons propagate in a diffusive manner and more slowly than their group velocity. Here, we resolve this controversy by means of molecular dynamics simulations of Rhodamine molecules in a Fabry-Pérot cavity. Our results suggest that polariton propagation is limited by the cavity lifetime and appears diffusive due to reversible population transfers between polaritonic states that propagate ballistically at their group velocity, and dark states that are stationary. Furthermore, because long-lived dark states transiently trap the excitation, propagation is observed on timescales beyond the intrinsic polariton lifetime. These insights not only help to better understand and interpret experimental observations, but also pave the way towards rational design of molecule-cavity systems for coherent exciton transport.

Solar cells based on organic molecules are promising alternatives to the silicon-based technologies that dominate today's market, mostly because organic photovoltaics (OPV) are cheaper to mass-produce, lighter, more flexible and easier to dispose of. A key step in light harvesting is transport of excitons from where photons are absorbed to where this energy is needed for initiating a photochemical process[1], usually deeper inside the material of the solar cell. Because excitons in organic materials are predominantly localized onto single molecules, exciton transport proceeds via incoherent hops[2]. Such random-walk diffusion is, however, too slow to compete with ultra-fast deactivation processes of singlet excitons, such as radiative and non-radiative decay. As exciton diffusion is furthermore hindered by thermal disorder, propagation distances in organic materials typically remain below 10 nm[2]. Such short diffusion lengths limit the efficiency of solar energy harvesting and require complex morphologies of active layers into nanometer sized domains, e.g., bulk heterojunctions in OPVs,

which not only complicates device fabrication, but also reduces device stability[3,4].

Distances of hundreds of nanometers have been observed for the diffusion of longer-lived triplet states[5], but because not all organic materials can undergo efficient intersystem crossing or singlet fission, it may be difficult to exploit triplet diffusion in general. Exciton mobility can also be increased through transient exciton delocalization[6-8], but as the direct excitonic interactions are weak in most organic materials, molecules need to be ordered to reach this enhanced transport regime.

Alternatively, permanent delocalization over large numbers of molecules can be achieved by strongly coupling excitons in the material to the confined light modes of optical cavities, such as Fabry-Pérot resonators (Fig. 1a) or nano-structured devices[9-11]. In this strong light-matter coupling regime the rate of energy exchange between molecular excitons and confined light modes exceeds the intrinsic

[1]Nanoscience Center and Department of Chemistry, University of Jyväskylä, P.O. Box 35, Jyväskylä 40014, Finland. [2]Departamento de Física Teórica de la Materia Condensada and Condensed Matter Physics Center (IFIMAC), Universidad Autónoma de Madrid, Madrid, Spain. [3]These authors contributed equally: Ilia Sokolovskii, Ruth H. Tichauer. ✉e-mail: gerrit.x.groenhof@jyu.fi

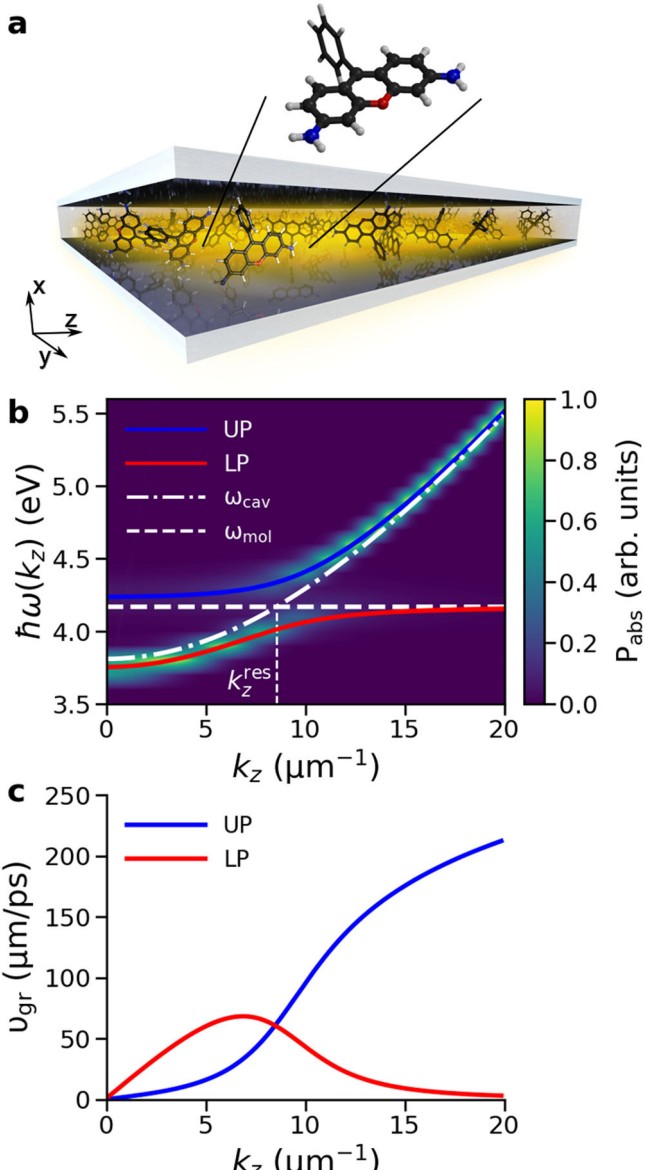

**Fig. 1 | Rhodamine-cavity system. a** Schematic illustration of an optical Fabry-Pérot microcavity filled with Rhodamine chromophores (not to scale). The quantum mechanical (QM) subsystem, shown in ball-and-stick representation in the inset, is described at the Hartree-Fock (HF/3-21G) level of theory in the electronic ground state ($S_0$), and at the Configuration Interaction level of theory, truncated at single electron excitations (CIS/3-21G), in the first singlet excited state ($S_1$). The molecular mechanical (MM) subsystem, consisting of the atoms shown in stick representation and the water molecules (not shown), is modeled with the Amber03 force field. **b** Normalized angle-resolved absorption spectrum ($P_{abs}$) of the cavity, showing Rabi splitting between lower polariton (LP, red line) and upper polariton (UP, blue line) branches. The cavity dispersion and excitation energy of the molecules (4.18 eV at the CIS/3-21G//Amber03 level of theory) are plotted with dot-dashed and dashed white lines, respectively. **c** Group velocity of the LP (red) and UP (blue), defined as the derivative of the frequency of polaritons $\omega(k_z)$ with respect to the in-plane wave-vector $k_z$, i.e., $\partial\omega(k_z)/\partial k_z$.

decay rates of both the excitons and the confined modes leading to the formation of new coherent light-matter states, called polaritons[12–20].

The majority of hybrid states in realistic molecule-cavity systems are dark[21–23], meaning that they have negligible contributions from the cavity photons. In contrast, the few states with such contributions are the bright polaritonic states that have dispersion and hence group velocity, defined as the derivative of the polariton energy with respect to in-plane momentum (i.e., $k_z$ in Fig. 1b). In the out-of-plane cavity direction (i.e., perpendicular to the mirrors), these states are delocalized over the molecules inside the mode volume, while in the in-plane direction (i.e., parallel to the mirrors) they behave as quasi-particles with a low effective mass and large group velocity (i.e., fractions of the speed of light). These polariton properties can be exploited for both out-of-plane[9–11,24–33], and in-plane energy transport[34–54].

Indeed, at cryogenic temperatures, in-plane ballistic propagation at the group velocity of polaritons was observed for polariton wave-packets in a Fabry-Pérot microcavity containing an $In_{0.05}Ga_{0.95}As$ quantum well[34]. Ballistic propagation was also observed for polaritons formed between organic molecules and Bloch surface waves[38,42,51], while a combination of ballistic transport on an ultrashort timescale (sub-50 fs) and diffusive motion on longer timescales was observed for cavity-free polaritons[43], for which strong coupling was achieved through a mismatch of the refractive indices between thin layers of densely-packed organic molecules and a host material[55]. In contrast, experiments on strongly coupled organic J-aggregates in metallic micro-cavities suggest that molecular polaritons propagate in a diffusive manner and much more slowly than their group velocities[40]. Furthermore, despite a low cavity lifetime on the order of tens of femtoseconds in these experiments, propagation was observed over several picoseconds, which was attributed to a long lifetime of the lower polariton (LP)[17,40]. Here, we address these controversies by providing atomistic insights into polariton propagation with multi-scale molecular dynamics (MD) simulations[56,57] of solvated Rhodamine molecules strongly coupled to the confined light modes of a one-dimensional (1D) Fabry-Pérot microcavity[58], shown schematically in Fig. 1a.

## Results and discussion
### Resonant excitation
First, we explore how polaritons propagate after resonant excitation of a Gaussian wavepacket of LP states with a broad-band laser pulse. In Fig. 2, we show the time evolution of the probability density of the polaritonic wave function, $|\Psi(t)|^2$ after such excitation in both a perfect lossless cavity with an infinite Q-factor ($\gamma_{cav} = 0\ ps^{-1}$, top panels) and a lossy cavity with a low Q-factor ($\gamma_{cav} = 66.7\ ps^{-1}$, bottom panels) containing 1024 Rhodamine molecules. Plots of wavepacket propagation in systems with 256 and 512 molecules are provided as Supplementary Information (SI, Figs. S5–S6), as well as animations of the wavepackets for all system sizes (Supplementary Movies 1–9 and 13–21).

**Lossless cavity.** In the perfect lossless cavity, the total wavepacket $|\Psi(t)|^2$ initially propagates ballistically close to the maximum group velocity of the LP branch ($v_{gr}^{LP,max} = 68\ \mu mps^{-1}$, Fig. 1c), until around 100 fs (see animations in the SI), when it slows down as evidenced by a decrease in the slope of the expectation value of the position of the wavepacket $\langle z \rangle$ in Fig. 3a. The change from a quadratic to a linear time-dependence of the Mean Squared Displacement (Fig. 3c) at $t = 100$ fs furthermore suggests a transition from ballistic to diffusive motion.

During propagation, the wavepacket broadens and sharp features appear, visible as vertical lines in both the total and molecular wave-packets in Fig. 2a, b and as peaks in the wavepacket animations provided as SI. These peaks coincide with the $z$ positions of molecules that contribute to the wavepacket with their excitations during propagation. Such peaks are not observed if there is no disorder and the molecular degrees of freedom are frozen (Fig. S16), but appear already at the start of the simulation when the initial configurations of the molecules are all different (Fig. S22). Similar observations were made by Agranovich and Gartstein[35] who attributed these peaks to energetic disorder among the molecular excitons. We therefore also assign these peaks to a partial localization of the wavepacket at the molecules due to structural disorder that alters their contribution to the wavepacket. In contrast, because the cavity modes are delocalized in space, the

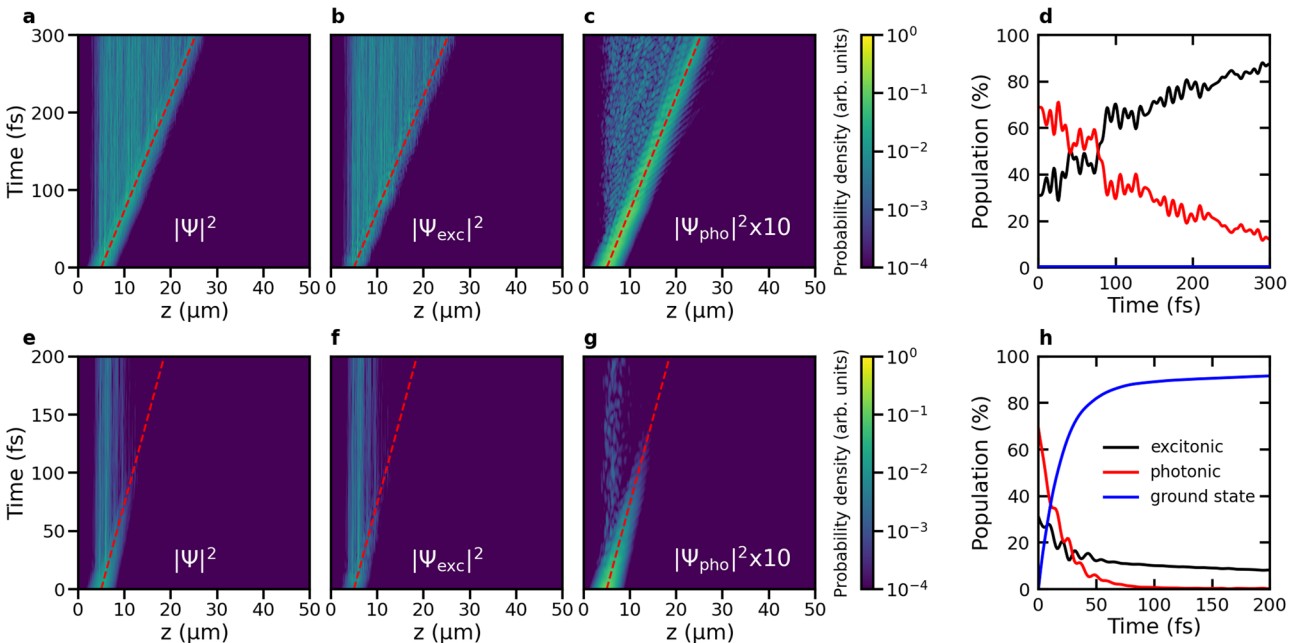

**Fig. 2 | Polariton propagation after resonant excitation of a wavepacket in the lower polariton (LP) branch centered at $z = 5\,\mu m$. a**, **b**, and **c**: total probability density $|\Psi(z, t)|^2$, probability density of the molecular excitons $|\Psi_{exc}(z, t)|^2$ and of the cavity mode excitations $|\Psi_{pho}(z, t)|^2$, respectively, as a function of distance ($z$, horizontal axis) and time (vertical axis), in a cavity with perfect mirrors (i.e., $\gamma_{cav} = 0\,ps^{-1}$). The red dashed line indicates propagation at the maximum group velocity of the LP ($68\,\mu mps^{-1}$). **d** Contributions of molecular excitons (black) and cavity mode excitations (red) to $|\Psi(z, t)|^2$ as a function of time in the perfect cavity. Without cavity losses, no ground state population (blue) can build up. **e–g** $|\Psi(z, t)|^2$, $|\Psi_{exc}(z, t)|^2$ and $|\Psi_{pho}(z, t)|^2$, respectively, as a function of distance ($z$, horizontal axis) and time (vertical axis), in a lossy cavity ($\gamma_{cav} = 66.7\,ps^{-1}$). **h** Contributions of the molecular excitons (black) and cavity mode excitations (red) to $|\Psi(z, t)|^2$ as a function of time. The population in the ground state, created by radiative decay through the imperfect mirrors, is plotted in blue. Source data are provided as a Source Data file.

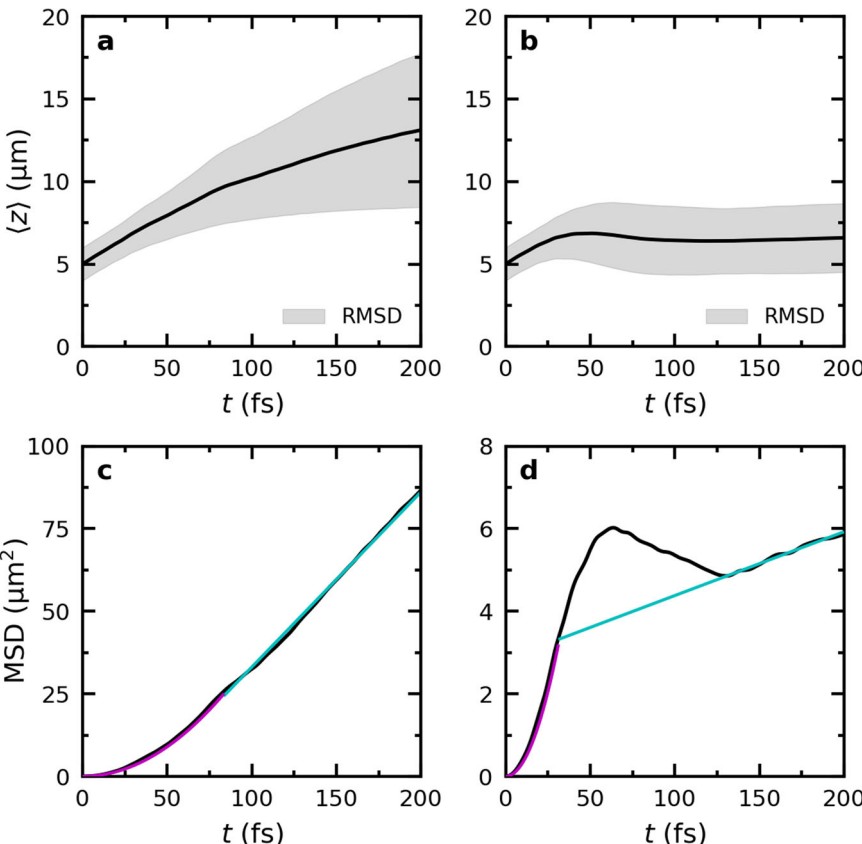

**Fig. 3 | Propagation of the polaritonic wavepacket after on-resonant excitation.** Top panels: Expectation value of the position of the total time-dependent wavefunction, $\langle z \rangle = \langle \Psi(z,t)|\hat{z}|\Psi(z,t)\rangle / \langle \Psi(z,t)|\Psi(z,t)\rangle$, in an ideal cavity (**a**, $\gamma_{cav} = 0\,ps^{-1}$) and a lossy cavity (**b**, $\gamma_{cav} = 66.7\,ps^{-1}$). The black lines represent $\langle z \rangle$ while the shaded area around the lines represents the root mean squared deviation (RMSD, i.e., $\sqrt{\langle (z(t) - \langle z(t) \rangle)^2 \rangle}$). Bottom panels: Mean squared displacement (MSD, i.e., $\langle (z(t) - \langle z(0) \rangle)^2 \rangle$) in the ideal (**c**) and the lossy (**d**) cavity. Magenta lines are quadratic fits to the MSD and cyan lines are linear fits. Source data are provided as a Source Data file.

photonic wavepacket remains smooth throughout the propagation (Fig. 2c).

The transition from ballistic propagation to diffusion around 100 fs coincides with the onset of the molecular excitons dominating the polaritonic wavepacket, as shown in Fig. 2d, in which we plot the contributions of the molecular excitons (black line) and cavity mode excitations (red line) to the total wave function (see Methods for details of this analysis). Because in the perfect cavity, photon leakage through the mirrors is absent (i.e., $\gamma_{cav} = 0$ ps$^{-1}$), the decrease of cavity mode excitations is due to population transfer from bright LP states into the dark state manifold (Fig. S20b)[59–61]. Thus, while resonant excitation of LP states initially leads to ballistic motion with the central group velocity of the wavepacket, as evidenced by the quadratic dependence of the Mean Squared Displacement on time (Fig. 3c), population transfer into dark states turns the propagation into a diffusion process, as evidenced by a linear time-dependence of the Mean Squared Displacement after ~100 fs.

Since dark states lack group velocity, and are therefore stationary, while excitonic couplings between molecules are neglected in our model (see SI), propagation in the diffusive regime must still involve bright polariton states. Our simulations therefore suggest that while, initially, molecular vibrations drive population transfer from the propagating bright states into the stationary dark states[62], this process is reversible, causing new wavepackets to form continuously within the full range of LP group velocities. Likewise, the propagation of transiently occupied bright states is continuously interrupted by transfers into dark states, and re-started with different group velocities. This re-spawning process leads to the diffusive propagation of the excitation observed in Fig. 2, with an increasing wavepacket width (Fig. 3a), in line with experimental observations[40,42,43,51].

**Lossy cavity.** Including a competing radiative decay channel by adding photon losses through the cavity mirrors at a rate of $\gamma_{cav} = 66.7$ ps$^{-1}$, leads to a rapid depletion of the polariton population (Fig. 2h), but does not affect the overall transport mechanism: the wavepacket still propagates in two phases, with a fast ballistic regime followed by slower diffusion. However, in contrast to the propagation in the ideal lossless cavity, we observe that the wavepacket temporarily contracts. This contraction is visible as a reduction of both the expectation value of $\langle z \rangle$ and the Mean Squared Displacement between 60 to 130 fs in the right panels of Fig. 3.

Initially, the propagation of the wavepacket is dominated by ballistic motion of the population in the bright polaritonic states moving at the maximum group velocity of the LP branch. However, due to non-adiabatic coupling[62], some of that population is transferred into dark states that are stationary. Because non-adiabatic population transfer is reversible, the wavepacket propagation undergoes a transition into a diffusion regime, which is significantly slower, as also observed in the ideal cavity (Fig. 3c).

In addition to these non-adiabatic transitions, radiative decay further depletes population from the propagating bright polaritonic states. Because before decay, this population has moved much further than the population that got trapped in the dark states, the expectation value of $\langle z \rangle$, as well as the Mean Squared Displacement, which were dominated initially by the fast-moving population, decrease until the slower diffusion process catches up and reaches the same distance around 130 fs (right panels in Fig. 3). Such contraction of the wavepacket in a lossy cavity is consistent with the measurements of Musser and co-workers, who also observe such contraction after on-resonant excitation of UP states[47].

Because of the contraction, it is difficult to see where the transition between the ballistic and diffusion regimes occurs in Fig. 3d. We therefore extrapolated the linear regime instead, and estimate the turn-over at 30 fs, where the quadratic fit to the ballistic regime intersects the extrapolated fit to the diffusion regime. As in the perfect lossless cavity, the transition between ballistic and diffusion regimes occurs when the population of molecular excitons exceeds the population of cavity mode excitations (Fig. 2h). However, due to the radiative decay of the latter, this turnover already happens around 30 fs in the lossy cavity simulations.

Owing to the short cavity mode lifetime (15 fs), most of the excitation has already decayed into the ground state at 100 fs, with a small remainder surviving in dark states (Fig. 2h) that lack mobility. Because cavity losses restrict the lifetime of bright LP states, the distance a wavepacket can reach is limited due to (i) a shortening of the ballistic phase, and (ii) a reduction of the diffusion coefficient (i.e., the slope of $\langle z \rangle$, Fig. 3b) in the second phase. Therefore, the overall velocity is significantly lower than in the perfect cavity, suggesting a connection between cavity Q-factor and propagation velocity[47], while also the broadening of the wavepacket is reduced (Fig. 3b). Furthermore, because the rate of population transfer is inversely proportional to the energy gap[62], and hence highest when the LP and dark states overlap[60], we speculate that the turn-over between the ballistic and diffusion regimes depends on the overlap between the absorption line width of the molecules and the polaritonic branches, and can hence be controlled by tuning the excitation energy to move the center of the initial polaritonic wavepacket along the LP branch. In addition, the direction of ballistic propagation can be controlled by varying the incidence angle of the on-resonant excitation pulse.

**Comparison to experiments.** Our observations are in line with transient microscopy experiments, in which broad-band excitation pulses were used to initiate polariton propagation. At low temperatures Freixanet et al. observed ballistic wavepacket propagation for a strongly coupled quantum dot[34]. If we suppress vibrations that drive population transfer by freezing the nuclear degrees of freedom, we also observe such purely ballistic motion (Figs. S16–S17). In contrast, in room temperature experiments on cavity-free molecular polaritons, Pandya et al.[43] identified two transport regimes: a short ballistic phase followed by diffusion. Based on the results of our simulations, we attribute the first phase to purely ballistic wavepacket propagation of photo-excited LP states. The slow-down of transport in the second phase is attributed to reversible trapping of population inside the stationary dark state manifold. Owing to the reversible transfer of population between these dark states and the LP states, propagation continues diffusively at time scales exceeding the polariton lifetime, in line with experiment[40,43].

### Off-resonant excitation

Next, we investigate polariton propagation after an off-resonant excitation of the molecule-cavity system. Experimentally such off-resonant excitation conditions are achieved by optically pumping a higher-energy electronic state of the molecules[38,40,42,51], which then rapidly relaxes into the lowest energy excited state (S$_1$) according to Kasha's rule[63]. We therefore modeled off-resonant photo-excitation by starting the simulations directly in the S$_1$ state of a single molecule, located at $z = 5$ μm in the cavity (SI). In Fig. 4, we show the time evolution of the probability density of the total polaritonic wave function, $|\Psi(t)|^2$, after such excitation in both a perfect lossless cavity with an infinite Q-factor ($\gamma_{cav} = 0$ ps$^{-1}$, top panels) and a lossy cavity with a low Q-factor ($\gamma_{cav} = 66.7$ ps$^{-1}$, bottom panels) containing 1024 Rhodamine molecules. Plots of the wavepacket propagation in systems with 256 and 512 molecules are provided as SI (Figs. S9–S10), as well as animations of the wavepackets for all system sizes (Supplementary Movies 25–33 and 37–45).

**Lossless cavity.** In the lossless cavity with perfect mirrors, the excitation, initially localized at a single molecule, rapidly spreads to other molecules (see animation in the SI). In contrast to the ballistic movement observed for on-resonant excitation, the wavepacket spreads out

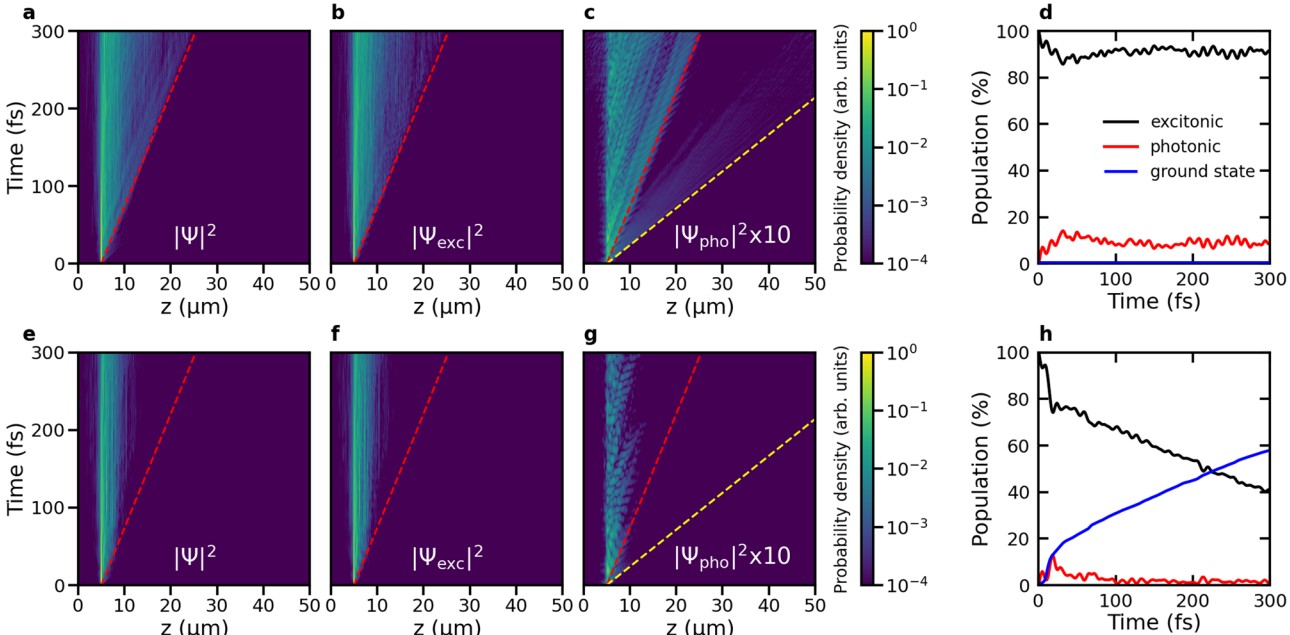

**Fig. 4 | Polariton propagation after off-resonant excitation into the $S_1$ state of a single molecule located at $z = 5\,\mu m$. a–c:** Total probability density $|\Psi(z, t)|^2$, probability density of the molecular excitons $|\Psi_{exc}(z, t)|^2$ and of the cavity mode excitations $|\Psi_{pho}(z, t)|^2$, respectively, as a function of distance ($z$, horizontal axis) and time (vertical axis), in a cavity with perfect mirrors (i.e., $\gamma_{cav} = 0\,ps^{-1}$). The red and yellow dashed lines indicate propagation at the maximum group velocity of the lower polaritons ($68\,\mu mps^{-1}$) and upper polaritons ($212\,\mu mps^{-1}$), respectively. **d** Contributions of the molecular excitons (black) and cavity mode excitations (red) to $|\Psi(z, t)|^2$ as a function of time in the perfect cavity. Without cavity decay, there is no build-up of ground state population (blue). **e–g** $|\Psi(z, t)|^2$, $|\Psi_{exc}(z, t)|^2$ and $|\Psi_{pho}(z, t)|^2$, respectively, as a function of distance ($z$, horizontal axis) and time (vertical axis), in a lossy cavity (i.e., $\gamma_{cav} = 66.7\,ps^{-1}$). **h** Contributions of the molecular excitons (black), and cavity mode excitations (red) to $|\Psi(z, t)|^2$ as a function of time in the lossy cavity. The population in the ground state, created by radiative decay through the imperfect mirrors, is plotted in blue. Source data are provided as a Source Data file.

instead, with the front of the wavepacket propagating at a velocity that closely matches the maximum group velocity of the LP branch ($68\,\mu mps^{-1}$, Fig. 1c), while the expectation value of the wavepacket position ($\langle z \rangle$, Fig. 5a) moves at a lower pace ($-10\,\mu mps^{-1}$).

Because we do not include negative $k_z$-vectors in our cavity model, propagation can only occur in the positive $z$ direction. With negative $k_z$-vectors, propagation in the opposite direction cancels such motion leading to $\langle z \rangle \approx 0$ (Fig. S15a). Nevertheless, since the Mean Squared Displacement is not affected by breaking the symmetry of the 1D cavity, and increases linearly with time in both uni- and bi-directional cavities (Figs. 5c and S15b), we consider it reasonable to assume that the mechanism underlying the propagation process is identical.

Because the population of dark states dominates throughout these simulations (Fig. 4d), and direct excitonic couplings are not accounted for in our model (SI), the observed propagation must again involve bright polariton states. Since the initial state, with one molecule excited, is not an eigenstate of the molecule-cavity system, population exchange from this state into the propagating bright states is not only due to displacements along vibrational modes that are overlapping with the non-adiabatic coupling vector[62], but also due to Rabi oscillations, in particular at the start of the simulation.

To quantify to what extent the overall propagation is driven by population transfers due to the molecular displacements, we performed additional simulations at 0 K with all nuclear degrees of freedom frozen. As shown in Fig. S18, the propagation is reduced at 0 K, and the wavepacket remains more localized on the molecule that was initially excited, than at 300 K. A quadratic time-dependence of the Mean Squared Displacement of the cavity mode contributions to the wavepacket (Fig. S19f) furthermore suggest that the mobility at 0 K is driven by the constructive and destructive interferences of the bright polaritonic states, which evolve with different phases (i.e., $e^{-iE_m t/\hbar}$).

The reduced mobility of the wavepacket at 0 K compared to 300 K (Fig. S19) confirms that thermally activated displacements of nuclear coordinates, which are absent at 0 K, are essential to drive population into the bright states and sustain the propagation of the polariton wavepacket. Thus, as during the diffusion phase observed for on-resonant excitation, ballistic motion of bright states is continuously interrupted and restarted with different group velocities, which makes the overall propagation appear diffusive with a Mean Squared Displacement that depends linearly on time (Fig. 5c), in line with experimental observations[40,42].

In the perfect cavity, propagation and broadening continue indefinitely due to the long-range ballistic motion of states with higher group velocities. Indeed, a small fraction at the front of the wavepacket, which moves even faster than the maximum group velocity of the LP (indicated by a yellow dashed line in Fig. 4c), is mostly composed of higher-energy UP states. These states not only have the highest in-plane momenta, but also relax most slowly into the dark state manifold of the perfect cavity due to the inverse dependence of the non-adiabatic coupling on the energy gap[57]. Momentum-resolved photo-luminenscence spectra at two distances from the initial excitation spot (Fig. S24) confirm that the front of the wavepacket is indeed composed of UP states: at short distances ($z = 10\,\mu m$) from the excitation spot ($z = 5\,\mu m$), the emission spectrum, accumulated over 100 fs simulation time, closely matches the full polariton dispersion of Fig. 1b, displaying both the LP and UP branches. In contrast, further away from the excitation spot ($z = 20\,\mu m$), the emission exclusively originates from the higher energy UP states, suggesting that only these states can reach the longer distance within 100 fs.

**Lossy cavity.** Adding a radiative decay channel for the cavity mode excitations ($\gamma_{cav} = 66.7\,ps^{-1}$) restricts the distance over which polaritons propagate (Fig. 4e–g), but does not affect the overall transport mechanism, as we also observe a linear increase of the Mean Squared

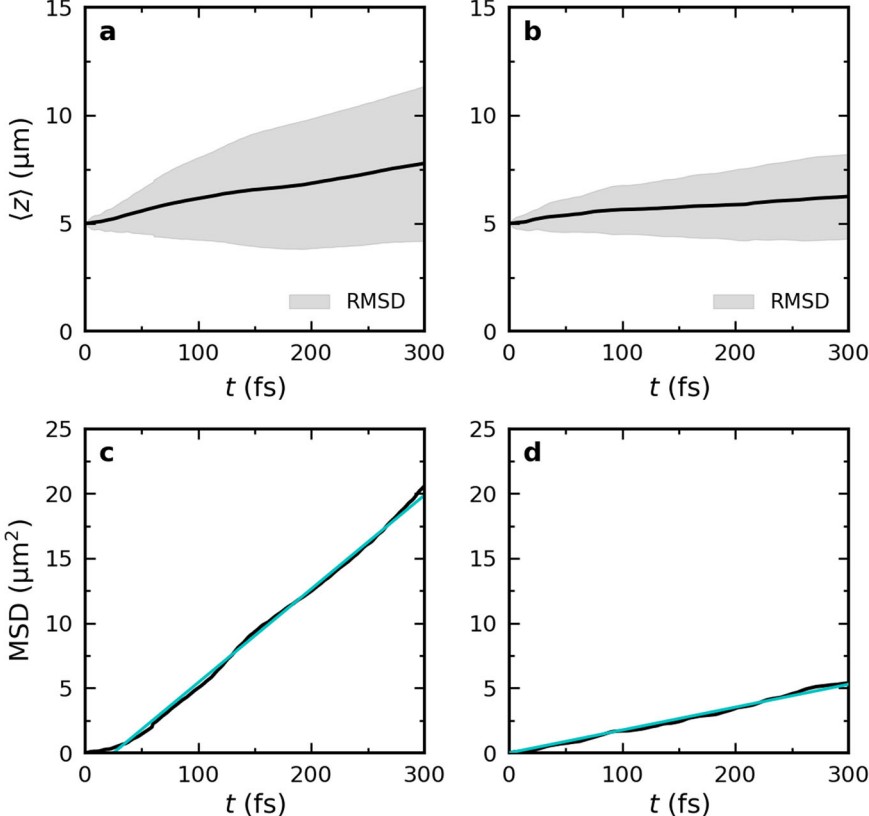

**Fig. 5 | Propagation of the polaritonic wavepacket after off-resonant excitation.** Top panels: Expectation value of the position of the total time-dependent wavefunction, $\langle z \rangle = \langle \Psi(z,t)|\hat{z}|\Psi(z,t)\rangle / \langle \Psi(z,t)|\Psi(z,t)\rangle$, in an ideal cavity (**a**, $\gamma_{cav} = 0$ ps$^{-1}$) and a lossy cavity (**b**, $\gamma_{cav} = 66.7$ ps$^{-1}$). The black lines represent $\langle z \rangle$ while the shaded area around the lines represents the root mean squared deviation (RMSD, i.e., $\sqrt{\langle (z(t) - \langle z(t)\rangle)^2 \rangle}$). Bottom panels: Mean squared displacement (MSD, i.e., $\langle (z(t) - z(0))^2 \rangle$) in the ideal lossless cavity (**c**) and the lossy cavity (**d**). Cyan lines are linear fits to the MSD. Source data are provided as a Source Data file.

Displacement with time (Fig. 5d). While the propagation in the lossy cavity initially is very similar to that in the ideal lossless cavity, radiative decay selectively depletes population from the propagating bright states and the wavepacket slows down, as evidenced by the expectation value of the displacement, $\langle z \rangle$, levelling off in Fig. 5b. In addition, since the maximum distance a wavepacket can travel in a lossy cavity is determined by the cavity lifetime in combination with the group velocity[38], the broadening of the wavepacket is also more limited when cavity losses are included (Fig. 5b). Furthermore, even if dark states do not have a significant contribution from the cavity mode excitations, the reversible transfer of population between the dark state manifold and the decaying bright polaritonic states, also leads to a significant reduction of dark state population in the lossy cavity as compared to the ideal lossless cavity (Fig. 4d, h). Nevertheless, dark states still provide protection from cavity losses as the overall lifetime of the photo-excited molecule-cavity system (>150 fs) significantly exceeds that of the cavity modes (15 fs).

**Comparison to experiments.** In microscopy experiments relying on off-resonant optical pumping, polariton emission is typically observed between the excitation spot and a point several microns further away[38,40,42,46,51]. While such a broad emission pattern is reminiscent of a diffusion process, the match between the total distance over which that emission is detected on the one hand, and the product of the maximum LP group velocity and cavity lifetime on the other hand, suggests ballistic propagation. The results of our simulations are thus in qualitative agreement with such observations as also our results suggest that, while polariton propagation appears diffusive under off-resonant excitation conditions, the front of the wave packet propagates close to the maximum group velocity of the LP branch.

Based on the analysis of our MD trajectories we propose that on the experimentally accessible timescales, polariton propagation appears diffusive due to reversible population transfers between stationary dark states and propagating bright states. For lossy cavities, radiative decay of the cavity modes further slows down polariton transport such that the excitation reaches a maximum distance before decaying completely. Because a large fraction of the population resides in the non-decaying dark states, the lifetime of the molecule-cavity system is extended[60], and polariton propagation can be observed on timescales far beyond the cavity lifetime, in line with experiment[40].

Note that in our simulations we only couple excitons to the modes of the Fabry-Pérot cavity, whereas in experiments with micro-cavities constituted by metallic mirrors, excitons can in principle also couple to surface plasmon polaritons (SPPs) below the light line that are supported by these metal surfaces. While their role will depend on the details of the set-up (e.g., the materials used, energy of the relevant molecular excitations, etc.), we cannot rule out that reversible population transfer between dark states and SPP-exciton polaritons also contributes to the effective diffusion constant observed in those experiments[40,48]. However, because SPPs decay exponentially away from the metal surface, and SPP-exciton polaritons also have group velocity, the qualitative behavior is not expected to change.

## Size dependence
The Rabi splitting depends on the vacuum field strength, $\mathbf{E}_y$, and the number of molecules, $N$, via $\hbar\Omega^{Rabi} \approx 2\boldsymbol{\mu}^{TDM} \cdot \mathbf{E}_y \sqrt{N}$, with $\boldsymbol{\mu}^{TDM}$ the molecular transition dipole moment, which for organic molecules is on the order of a few Debye. Because the vacuum field strength of a cavity is inversely proportional to the square root of the mode volume (Eq. 3

in SI), the Rabi splitting scales with the molecular concentration in the mode volume, $V_{cav}$, of the cavity, i.e., $\hbar\Omega^{Rabi} \propto \sqrt{N/V_{cav}}$. Reaching the strong coupling regime to form polaritons with organic molecules in Fabry-Pérot cavities with mode volumes on the order of $V_{cav} \propto (\lambda_{cav}/n)^3$ (where $\lambda_{cav}$ is the wavelength of the cavity mode and $n$ the refractive index), thus requires collective coupling of many molecules (i.e., $10^5$–$10^8$)[64–66]. Because the number of molecules we can include in our simulations is much smaller due to limitations on hard- and software, we investigated how that number affects the propagation by repeating simulations for different $N$. To keep the Rabi splitting constant, and hence the polariton dispersion the same, we scaled the mode volume with $N$, i.e., $V_{cav} = NV_{cav,0}$, where $V_{cav,0}$ is the mode volume required to achieve a Rabi splitting of 325 meV with a single Rhodamine molecule in the cavity.

With the exception of the smallest ensemble that lacks dark states, we observe for all other ensemble sizes that the propagation mechanism involves reversible population exchange between the stationary dark state manifold and propagating polaritons (Figs. S5–S13, SI). These additional simulations therefore underscore the role of dark states in the propagation process and suggest that the mechanism does not strongly depend on $N$. In contrast to the mechanism, however, the rates at which these population exchanges occur, depend on the number of molecules. Indeed, these rates are inversely proportional to $N$[23,62,67]. Because the number of dark states scales with $N$, whereas the number of polaritonic states is constant (Fig. S4), we observe that for the larger ensembles, the fraction of population residing within the dark state manifold is higher than for the smaller ensembles. Such differences affect (i) the propagation velocity (e.g., Fig. S14); (ii) the lifetime (e.g., right columns in Figs. 4 and S11) and therefore also (iii) the distance over which the exciton-polaritons are transferred (e.g., Figs. 5, and S11–S13).

Because the velocity is inversely proportional to $N$ (Fig. S14), the propagation velocity in experiments, with $10^5$–$10^8$ molecules inside the mode volume[64–66], is much lower than in our simulations. Nevertheless, because of the $1/N$ scaling, the effective polariton propagation velocity approaches the lower experimental limit of $10^5$ coupled molecules[64] already around 1000 molecules. We therefore consider the results of the simulations with 1024 Rhodamines sufficiently representative for experiment and for providing qualitative insights into polariton propagation. Indeed, a propagation speed of 9.6 µmps⁻¹ in the cavity containing 1024 molecules is about an order of magnitude below the maximum group velocity of the LP (68 µmps⁻¹) in line with experiments on organic microcavities[40], and cavity-free polaritons[43].

## Summary and outlook

To conclude, we have investigated exciton transport in cavities filled with Rhodamine molecules by means of atomistic MD simulations that not only include the details of the cavity mode structure[57,58], but also the chemical details of the material[56]. The results of our simulations suggest that transport is driven by an interplay between propagating bright polaritonic states and stationary dark states. Reversible population exchanges between these states interrupt ballistic motion in bright states and make the overall propagation process appear diffusive. While for off-resonant excitation of the molecule-cavity system, these exchanges are essential to transfer population from the initially excited molecule into the bright polaritonic branches and start the propagation process, the exchanges limit the duration of the initial ballistic phase for on-resonant excitation. As radiative decay of the cavity modes selectively depletes the population in bright states, ballistic propagation is restricted even further if the cavity is lossy. Because dark states lack in-plane momentum, the reversible population exchange between dark and bright states causes diffusion in all directions. Therefore, under off-resonant excitation conditions, the propagation direction cannot be controlled. In contrast, because bright states carry momentum, the propagation direction in

the ballistic phase can be controlled precisely by tuning the incidence angle and excitation wavelength under on-resonant excitation conditions.

The rate at which population transfers between bright and dark states depends on the non-adiabatic coupling vector, whose direction and magnitude are determined by the Huang-Rhys factor in combination with the frequency of the Franck-Condon active vibrations[62], both of which are related to the molecular Stokes shift[68]. In addition, because the non-adiabatic coupling is inversely proportional to the energy gap[62], the Stokes shift in combination with the Rabi splitting, also determines the region on the LP branch into which population transfers after off-resonant excitation of a single molecule[69–72]. We therefore speculate that the Stokes shift can be an important control knob for tuning the coherent propagation of polaritons.

Because our Rhodamine model features the key photophysical characteristics of an organic dye molecule, we speculate that the propagation mechanism observed in our simulations is generally valid for exciton transport in strongly-coupled organic micro-cavities, in which the absorption line width of the material exceeds the Rabi splitting and there is a significant overlap between bright and dark states. To confirm this, we have also performed simulations of exciton transport in cavities containing Tetracene and Methylene Blue and observed that the propagation mechanism remains the same (Figs. S25–S30, Supplementary Movies 10–12 and 46–48). Future work will be aimed at investigating how the propagation can be controlled by tuning molecular parameters, temperature, Rabi splitting (Fig. S23), or cavity Q-factor[73]. Because we include the structural details of both cavity and molecules, our simulations, which are in qualitative agreement with experiments, could be used to systematically optimize molecule-cavity systems for enhancing energy transfer.

## Methods

### Multiscale Tavis-Cummings simulation model

We used the multi-scale Tavis-Cummings model, introduced by Luk et al.[56], and extended to the multiple modes of a one-dimensional (1D) Fabry-Pérot micro-cavity[58] by Tichauer et al.[57], to perform molecular dynamics (MD) simulations of 1024 solvated Rhodamine molecules strongly coupled to the confined light modes of a 1D Fabry-Pérot micro-cavity, shown in Fig. S1[58]. In this model, we apply the Born-Oppenheimer approximation to separate the nuclear degrees of freedom, which we treat classically, from the electronic degrees of freedom and the cavity modes. Within the single-excitation subspace, probed experimentally under weak driving conditions, and employing the rotating wave approximation (RWA), valid for light-matter coupling strengths below 10% of the material excitation energy[74], we model the electronic plus cavity mode degrees of freedom with the Tavis-Cummings model of Quantum Optics[75,76]. In the long-wavelength approximation, the interaction between the molecular excitons and the cavity modes are modeled as the inner products between the transition dipole moments and the vacuum field associated with an excitation of the Fabry-Pérot cavity modes. The multi-scale Tavis-Cummings model is described in our previous works[56,57,60], and we provide a concise summary of the details relevant to this work in Section 1 of the SI.

### Rhodamine model

The electronic ground state ($S_0$) of the Rhodamine molecules was modeled at the hybrid Quantum Mechanics / Molecular Mechanics (QM/MM) level[77,78], using the restricted Hartree-Fock (HF) method in combination with the 3-21G basis set[79] for the QM subsystem, which contains the fused rings of the molecule. The MM subsystem, consisting of the rest of the molecule and 3,684 TIP3P waters[80], was modeled with the Amber03 force field[81]. The first electronic excited state ($S_1$) of the QM region was modeled with Configuration Interaction, truncated at single electron excitations (CIS/3-21G//Amber03). At

this level of theory, the excitation energy of Rhodamine is 4.18 eV, which is significantly overestimated with respect to experiments. This discrepancy is due to the limited size of the basis set and the neglect of electron-electron correlation in the ab initio methods we used. While including electron-electron correlation into the description of the QM region improves the vertical excitation energy, we show in the SI that this does not significantly change the topology of the relevant potential energy surfaces (Fig. S3), which determines the molecular dynamics. Further details of the Rhodamine simulation setup, as well as the full details of additional simulations of Tetracene in cyclohexane and of Methylene Blue in water, are provided in the SI.

## Molecular dynamics of Rhodamine-cavity systems

**Cavity model.** After a 200 ns equilibration at the force field level, and a further 100 ps equilibration at the QM/MM level, the 1024 Rhodamine molecules were placed with equal inter-molecular distances on the $z$-axis of a periodic 1D cavity[35,58] of length $L_z = 50$ μm, where $z$ indicates the in-plane direction (i.e., parallel to the mirrors). With a distance of $L_x = 163$ nm between the mirrors (cavity width), where $x$ indicates the out-of-plane direction (i.e., perpendicular to the mirrors), the fundamental mode of the cavity has an energy of $\hbar\omega_0 = 3.81$ eV at normal incidence (i.e., $k_z = 0$) and hence its dispersion is red-detuned with respect to the molecular excitation energy at 4.18 eV (horizontal dashed white line in Fig. 1b). The dispersion, $\omega_{cav}(k_z) = \sqrt{\omega_0^2 + c^2 k_z^2/n^2}$ (dot-dashed white line in Fig. 1b), was modeled with 160 modes ($0 \leq p \leq 159$ for $k_z = 2\pi p/L_z$, with $c$ the speed of light and $n$ the refractive index)[35]. With a cavity vacuum field strength of 0.26 MVcm$^{-1}$, the Rabi splitting, defined as the energy difference between the bright lower (LP) and upper polariton (UP) branches at the wave-vector $k_z^{res}$ where the cavity dispersion matches the molecular excitation energy (Fig. 1b), was ~ 325 meV. While the choice for a 1D cavity model with only positive $k_z$ vectors was motivated by the necessity to keep our simulations computationally tractable, it precludes the observation of elastic scattering events that would change the direction (i.e., in-plane momentum, $\hbar k$) of propagation. Furthermore, with only positive $k_z$ vectors, polariton motion is restricted to the + $z$ direction, but we show in the SI (Fig. S15) that this assumption does not affect our conclusions about the transport mechanism. To maximize the collective light-matter coupling strength, the transition dipole moments of the Rhodamine molecules were aligned to the vacuum field at the start of the simulation. The same starting coordinates were used for all Rhodamines, but different initial velocities were selected randomly from a Maxwell-Boltzmann distribution at 300 K. We checked that adding disorder by randomly selecting configurations from the equilibrium QM/MM trajectory, or by randomly placing molecules on the $z$-axis, does not affect the conclusions of our work (Figs. S21 and 22).

**Mean-field molecular dynamics.** Ehrenfest MD trajectories were computed by numerically integrating Newton's equations of motion using a leap-frog algorithm with a 0.1 fs timestep[82]. The multi-mode Tavis-Cummings Hamiltonian (Eq. 4 in SI) was diagonalized at each time-step to obtain the (adiabatic) polaritonic eigenstates[75,76]:

$$|\psi^m\rangle = \left(\sum_j^N \beta_j^m \hat{\sigma}_j^+ + \sum_p^{n_{mode}} \alpha_p^m \hat{a}_p^\dagger\right)\left|S_0^1 S_0^2 .. S_0^{N-1} S_0^N\right\rangle|0\rangle \quad (1)$$

with eigenenergies $E_m$. Here, $|S_0^1 S_0^2 .. S_0^{N-1} S_0^N\rangle|0\rangle$ represents the wave function of the molecule-cavity system in the ground state, in which neither the molecules, nor the cavity modes are excited. The creation operators $\hat{\sigma}_j^+ = |S_1^j\rangle\langle S_0^j|$ and $\hat{a}_p^\dagger = |1_p\rangle\langle 0_p|$ excite molecule $j$ and cavity mode $p$ with in-plane momentum $k_z = 2p\pi/L_z$, respectively. The $\beta_j^m$ and $\alpha_p^m$ expansion coefficients thus reflect the contribution of the molecular excitons ($|S_1^j\rangle$) and of the cavity mode excitations ($|1_p\rangle$) to polariton $|\psi^m\rangle$.

The total wavefunction, $|\Psi(t)\rangle$, was coherently propagated along with the classical degrees of freedom of the molecules as a time-dependent superposition of the polaritonic eigenstates:

$$|\Psi(t)\rangle = \sum_m c_m(t)|\psi^m\rangle \quad (2)$$

where $c_m(t)$ are the time-dependent expansion coefficients of the time-independent eigenstates $|\psi^m\rangle$. A unitary propagator in the local diabatic basis was used to integrate these coefficients[83], while the nuclear degrees of freedom of the molecules were evolved on the mean-field potential energy surface.

Results reported in this work were obtained as averages over at least two trajectories. For all simulations we used Gromacs 4.5.3[84], in which the multi-mode Tavis-Cummings QM/MM model was implemented[57], in combination with Gaussian16[85]. Further details of the simulations, including other ensemble sizes for the Rhodamine-cavity systems, and different molecules, i.e., Tetracene and Methylene Blue (Fig. S2), are provided in the SI.

**Excitation conditions.** Resonant excitation into the LP branch by a short broad-band laser pulse, as often used in time-resolved experiments[34,43,47], was modeled by preparing a Gaussian wavepacket of LP states centered at $\hbar\omega = 3.94$ eV where the group velocity of the LP branch is highest, and with a bandwidth of $\sigma = 0.707$ μm$^{-1}$ [35]. Thus, the expansion coefficients of the wave function, $|\Psi(t=0)\rangle$ (Eq. 2), were initiated as

$$c_m(t=0) = \left(\frac{\zeta}{2\pi^3}\right)^{\frac{1}{4}} \exp[-\zeta(k_z^m - k_c)^2] \quad (3)$$

with $\zeta = 10^{-12}$m$^2$ characterizing the width of the wavepacket and $k_z^m$ the expectation value of the in-plane momentum of polariton $|\psi_m\rangle$ (i.e., $\langle k_z^m \rangle = \sum_p^{n_{mode}} |\alpha_p^m|^2 k_{z,p} / \sum_p^{n_{mode}} |\alpha_p^m|^2$).

Experimentally, an off-resonant excitation in a molecule-cavity system is achieved by optically pumping a higher-energy electronic state of the molecules[38,40,42,51], which then rapidly relaxes into the lowest energy excited state (S$_1$) according to Kasha's rule[63]. We therefore modeled off-resonant photo-excitation by starting the simulations directly in the S$_1$ state of a single molecule, located at $z = 5$ μm in the cavity. This was achieved by initiating the expansion coefficients of the wave function (Eq. 2) as $c_m(t=0) = \beta_j^m$. A more detailed derivation of these initial conditions is provided in the SI.

We assume that the intensity of the excitation pulse in both cases is sufficiently weak for the system to remain within the single-excitation subspace in our simulations. We thus exclude multi-photon absorption and model the interaction with the pump pulse as an instantaneous absorption of a single photon.

**Cavity lifetime.** Because the light-confining structures used in previous experiments (e.g., Fabry-Pérot cavities[34,40,47,48], Bloch surface waves[38,42,51], or plasmonic lattices[41,46,54]) span a wide range of quality factors (Q-factors), we also investigated the effect of the cavity mode lifetime on the transport by performing simulations in an ideal lossless cavity with no photon decay (i.e., $\gamma_{cav} = 0$ ps$^{-1}$), and a lossy cavity with decay rate of 66.7 ps$^{-1}$. This decay rate corresponds to a lifetime of 15 fs, which is in the same order of magnitude as the 2–15 fs lifetimes reported for metallic Fabry-Pérot cavities in experiments[40,86–88]. Cavity losses were modeled as a first-order decay of population from eigenstates with contributions from cavity mode excitations. Assuming that the intrinsic decay rates $\gamma_{cav}$ are the same for all modes, the total loss rate of an eigenstate, $|\psi^m\rangle$, is calculated as the product of $\gamma_{cav}$ and the total photonic weight, $\sum_p |\alpha_p^m|^2$, of that eigenstate.

In addition to cavity loss, also internal conversion via the conical intersection seam between the $S_1$ and $S_0$ potential energy surfaces[89] can provide a decay channel for the excitation. However, because in our Rhodamine model the minimum energy conical intersection is 1.3 eV higher in energy than the vertical excitation (Supplementary Table 1 in SI) and is therefore unlikely to be reached on the timescale of our simulations, we neglect internal conversion processes altogether.

## Wavefunction analysis

To monitor the propagation of the wavepacket, we plotted the probability density of the total time-dependent wave function $|\Psi(t)|^2$ at the positions of the molecules, $z_j$, as a function of time (Figs. 2 and 4). We thus represent the density as a discrete distribution at grid points that correspond to the molecular positions, rather than a continuous distribution. In addition to the total probability density, $|\Psi(t)|^2$, we also plotted the probability densities of the excitonic $|\Psi_{\mathrm{exc}}(t)|^2$ and photonic $|\Psi_{\mathrm{pho}}(t)|^2$ contributions separately (b, f and c, g, respectively, of Figs. 2 and 4).

The amplitude of $\left|\Psi_{\mathrm{exc}}(t)\right\rangle$ at position $z_j$ in the 1D cavity (with $z_j = (j-1)L_z/N$ for $1 \leq j \leq N$) is obtained by projecting the excitonic basis state in which molecule $j$ at position $z_j$ is excited ($|\phi_j\rangle = \hat{\sigma}_j^+|\phi_0\rangle$), on the total wave function (Eq. 2):

$$
\begin{aligned}
\left|\Psi_{\mathrm{exc}}(z_j,t)\right\rangle &= \hat{\sigma}_j^+|\phi_0\rangle\langle\phi_0|\hat{\sigma}_j|\Psi(t)\rangle \\
&= \sum_m^{N+n_{\mathrm{mode}}} c_m(t)\beta_j^m \hat{\sigma}_j^+|\phi_0\rangle
\end{aligned}
\tag{4}
$$

where the $\beta_j^m$ are the expansion coefficients of the excitonic basis states in polaritonic state $|\psi^m\rangle$ (Eq. 1) and $c_m(t)$ the time-dependent expansion coefficients of the total wavefunction $|\Psi(t)\rangle$ (Eq. 2).

The cavity mode excitations are described as plane waves that are delocalized in real space. We therefore obtain the amplitude of the cavity modes in polaritonic eigenstate $|\psi^m\rangle$ at position $z_j$ by Fourier transforming the projection of the cavity mode excitation basis states, in which cavity mode $p$ is excited ($\left|\phi_p\right\rangle = \hat{a}_p^\dagger|\phi_0\rangle$):

$$
\begin{aligned}
\left|\psi_{\mathrm{pho}}^m(z_j)\right\rangle &= \mathcal{FT}^{-1}\left[\sum_p^{n_{\mathrm{mode}}} \hat{a}_p^\dagger|\phi_0\rangle\langle\phi_0|\hat{a}_p|\psi^m\rangle\right] \\
&= \frac{1}{\sqrt{N}}\sum_p^{n_{\mathrm{mode}}} \alpha_p^m e^{i2\pi z_j p}\hat{a}_p^\dagger|\phi_0\rangle
\end{aligned}
\tag{5}
$$

where the $\alpha_p^m$ are the expansion coefficients of the cavity mode excitations in polaritonic state $|\psi^m\rangle$ (Eq. 1) and we normalize by $1/\sqrt{N}$ rather than $1/\sqrt{L_z}$, as we represent the density on the grid of molecular positions. The total contribution of the cavity mode excitations to the wavepacket at position $z_j$ at time $t$ is then obtained as the weighted sum over the Fourier transforms:

$$
\begin{aligned}
\left|\Psi_{\mathrm{pho}}(z_j,t)\right\rangle &= \sum_m^{N+n_{\mathrm{mode}}} c_m(t)\times\mathcal{FT}^{-1}\left[\sum_p^{n_{\mathrm{mode}}} \hat{a}_p^\dagger|\phi_0\rangle\langle\phi_0|\hat{a}_p|\psi^m\rangle\right] \\
&= \sum_m^{N+n_{\mathrm{mode}}} c_m(t)\frac{1}{\sqrt{N}}\sum_p^{n_{\mathrm{mode}}} \alpha_p^m e^{i2\pi z_j p}\hat{a}_p^\dagger|\phi_0\rangle
\end{aligned}
\tag{6}
$$

with $c_m(t)$ the time-dependent expansion coefficients of the adiabatic polaritonic states $|\psi^m\rangle$ in the total wavefunction $|\Psi(t)\rangle$ (Eq. 2).

## Data availability

All data, including simulations models, input files, trajectories and structures, analysis scripts and programs, including raw data, are available for download from IDA - Research Data Storage[90]. Source data are provided with this paper.

## Code availability

The GROMACS-4.5.3 fork, in which the multi-scale Tavis-Cummings model used in this work, was implemented, is available for download from GitHub[91].

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

## Acknowledgements

This work was supported by the Academy of Finland (Grant No. 323996 and 332743 to G.G.), the European Research Council (Grant No. ERC-2016-StG-714870 to J.F.), and by the Spanish Ministry for Science, Innovation, Universities-Agencia Estatal de Investigación (AEI) to J.F. through Grants PID2021-125894NB-I00 and CEX2018-000805-M (through the María de Maeztu program for Units of Excellence in Research and Development). We thank J. Jussi Toppari, A. M. Berghuis, J. Gómez Rivas, T. Schwartz, M. Balusubrahmaniyam and D. Sanvitto for fruitful discussions. We also thank the Center for Scientific Computing (CSC-IT Center for Science) for generous computational resources, and N. Runenberg for his assistance in running the simulations on these resources.

## Author contributions

G.G. and J.F. acquired funding; I.S., R.H.T., and G.G. performed the Molecular Dynamics simulations; D.M. performed the Quantum Chemistry calculations; I.S., R.H.T., D.M., J.F., and G.G. performed data analysis; I.S., R.H.T., and G.G. designed the research; I.S., R.H.T., and G.G. wrote the paper, which all authors revised and edited. I.S. and R.H.T. contributed equally to this work. All authors approved the final version of the paper.

## Competing interests

The authors declare no competing interests.
