## [Peer Review File · Nature Communications]

Multi-scale molecular dynamics simulations of enhanced energy transfer in organic molecules under strong couplingREVIEWER COMMENTS

Reviewer #1 (Remarks to the Author):

The manuscript by Sokolovskii et. al. entitled, “Enhanced Excitation Energy Transfer under Strong Light-Matter Coupling: Insights from Multi-Scale Molecular Dynamics Simulations” describes an attempt to understand the excitation energy transfer between Rhodamine molecules under electronic strong coupling conditions. Authors detailed the transport dynamics of polaritons in lossless and lossy cavities upon on and off resonant excitation conditions. They suggest that the cavity lifetime and reversible population transfers between polaritonic states and dark states control the ballistic and diffusive propagation of the polaritons. The manuscript is nicely written however there are some major issues need to be checked before considering it for publication in Nat. Commun.

Light matter strong coupling and thus the polaritons are getting immense attention in different fields of science. The work would have significant impact towards the understanding of polaritonic dynamics, if the approximations are little more realistic.

Recommendation: Publish after a major revision noted

Following are the scientific questions / comments.

1. The authors previously also have reported various interesting studies with Rhodamine molecules inside an optical cavity. I have a strong concern about the basic computation characterization of the molecule. Experimentally Rhodamine molecules have S0 to S1 transition at around 500 nm (~2.5 eV). But in the present manuscripts as well as in the earlier manuscripts by the authors (J. Chem. Phys. 154, 104112, and J. Chem. Theory Comput. 13, 4324), the S0 to S1 transition in Rhodamine is at 4.18 eV. Why there is a huge difference in the energy values? Can it be improved?
2. The calculation in a lossy cavity is performed by considering the rate of cavity loss as 66.7 ps^{-1} (cavity photon life time ~15 fs). From where authors arrived at this value? In Page number 4, Column 1, Line 186, Reference 58 is cited. In that Ref., the photon lifetime in a lossy cavity is given as 25 fs.
3. Reported Q factor in most of the experiments is around 100. Can it be considered in the calculation, instead of cavity with infinity Q factor?
4. What is the Q factor of the lossy cavity?
5. In the lossless cavity, the cavity loss is absent, however what about the loss from the molecular decay? In the on-resonant conditions, after 100 fs, excitonic population is increasing. So, if molecular decay is considered, it should generate GS of the molecule. It would be nice to have some thoughts on it.
6. Polariton transport dynamics is reported by exciting the lower polariton (LP) and dark states (DS) in on resonant and off-resonant conditions respectively. When there is an efficient population transfer between the LP and DS, what is role of excitation energy?
7. In continuation with the comment 5, especially in the lossy cavity, why the dynamics is different upon on and off resonant excitation?
8. If I understand correct, dark states are superpositions of collective molecular excitations. Can authors comment on the ratio between the polaritonic states to Dark states in their

calculation?

9. Upon off resonant excitation to S1 state of the molecule, how the upper polariton is observed (Figure 4c)?

Additional minor suggestions,

1. In Page 2; Column 1, Line 92, It is written like “The majority of hybrid states in realistic molecule-cavity systems are dark, [20–22] meaning that they have no contributions from the cavity modes,” I think, dark states will have minor contribution from the cavity modes. We cannot say that there is no contribution.

2. The figure legends, molecular and GS are confusing in Fig. 2d, h and 3d, h. The term molecular could be changed to excitonic to have better clarity.

Reviewer #2 (Remarks to the Author):

The manuscript presents theoretical investigations into polariton-assisted exciton transport and is both timely and interesting. However, there are a few points that could be improved.

- Firstly, it would be helpful to clarify the role of the Stokes shift in resonant LP excitation (are conclusions specific to Rhodamines?).

- Secondly, I am curious to know if the authors have any comments on the guided exciton-polaritons that exist below the light line, and their role in enhanced exciton transport (it appears that the authors have only discussed exciton coupling to FP modes above the light line).

- Furthermore, with regards to LP->DS transfer, it would be valuable to explore the role of the Stokes shift, Rabi splitting and temperature in this process. It seems to me that it could have been beneficial to consider working with zero Stokes shift systems like J-aggregates, while maximizing the Rabi splitting.

- Finally, I believe there appears to be a typo in line 149 which should read "provide".

Overall, this manuscript deserves the attention of the scientific community.

Reviewer #3 (Remarks to the Author):

In this manuscript, Sokolovskii performed the multiscale MD simulations of exciton polariton propagation in a 1D Fabry-Perot cavity. Results with two different initial conditions were discussed: (i) exciting the LP mode at the maximal group velocity; (ii) exciting the S₁ state of a single molecule. Their results showed that the LP propagation along the cavity mirror plane is initially ballistic and becomes diffusive in later times. They further analyzed the origin of this diffusive behavior and studied the important roles of the molecular dark modes. As the first atomistic simulation of exciton polariton propagation along the cavity mirror plane, this manuscript is likely to attract a broad audience of Nature Communications. Hence, I would recommend the publication of this manuscript if the following points are properly addressed.

1. According to the theory of multi-scale MD for exciton-polaritons, only the first excited states of molecules and photons are included. Upon exciting the polariton or S₁ state, this approximation is valid if the time duration is short. In later times (e.g., the diffusive motion of polaritons), however, the total wavefunction of all molecules + photons could contain double excitations (e.g., two molecules in the first excited states), which is not described within this framework. Could the authors estimate the effects of the missing double excitations in their simulations?
2. One largest issue on simulating polaritons is the so-called “large N problem”: because it is very hard to explicitly simulate millions (the experimental value) of chromophores, one can only simulate a finite (usually tens to thousands) number of chromophores and then check the system size dependence of the observables by fixing the Rabi splitting. The macroscopic limit, or the large N limit, should correspond to the experiments. The authors did this system size check in the SI, but in the main text it is relatively hard to find the N-dependence of many interesting observations, especially the reversible energy transfer from dark modes to polaritons in the resonant excitation case (Figs. 2 and 3) and the off-resonant excitation case (Fig. 4 and 5). Hence, I strongly suggest the authors to briefly discuss the N-dependence for their observations in the main text: what the results would be if N approaches the experimental limit?
3. Relating to the above comment, in page 5, the authors stated, “*This re-spawning process leads to the diffusive propagation of the excitation observed in Figure 2, with an increasing wavepacket width (Figure 3a), in line with experimental observations [37, 39, 40, 45].*” I would assume that the re-spawning process would become very weak when N is large, no?
4. In page 7, the authors stated, “*Because the overall propagation velocity is determined by the population in the bright states, and that population is inversely proportional to N due to the inverse scaling of the non-adiabatic population transfer rate [62], also the velocity is inversely proportional to N (Figure S12).*” In the SI, the authors fitted the N dependent results of the average propagation velocity v_{av} with $v_{av} = \frac{1}{N} + v_{N \rightarrow \infty}$ and claimed using 1000 molecules is good enough to get the macroscopic limit. Why do the authors not use

the formula $v_{av} = \frac{1}{N}$ to do the fitting? Of course, with this new formula one would get zero v_{av} in the macroscopic limit (which is not good to compare with experiments). I would assume that if only a single molecule is initially excited, no polariton excitation would be observed at all in the experimental limit. In experiments, the off-resonant excitation probably excites a finite fraction of molecules instead of a single molecule, so the enhanced diffusion could be observed in experiments.

5. In page 4, the authors stated, “*Because no peaks are observed if the molecular degrees of freedom are frozen (Figure S14), we infer that these localizations are a manifestation of energetic disorder among the molecules that affects their contribution to the wavepacket [32].*” From my understanding, Fig. S14 seems to indicate that in the absence of thermal motion, polaritons would not dephase to the dark modes, while Fig. 2 appears to show that with thermal motion, polaritons can dephase to the dark modes. Such a dephasing process is of course temperature dependent. I cannot understand why it is related to the localization of polaritons due to energy disorder.

Overall, this manuscript is of high quality, and reading it is a delightful experience.

Reviewer 1

General comment 1: The manuscript by Sokolovskii et. al. entitled, “Enhanced Excitation Energy Transfer under Strong Light-Matter Coupling: Insights from Multi-Scale Molecular Dynamics Simulations” describes an attempt to understand the excitation energy transfer between Rhodamine molecules under electronic strong coupling conditions. Authors detailed the transport dynamics of polaritons in lossless and lossy cavities upon on and off resonant excitation conditions. They suggest that the cavity lifetime and reversible population transfers between polaritonic states and dark states control the ballistic and diffusive propagation of the polaritons. The manuscript is nicely written however there are some major issues need to be checked before considering it for publication in Nat. Commun.

General response 1: We thank the reviewer for the time and effort spent in evaluating our manuscript. We agree with the concise summary of our work and are pleased that the reviewer considers our manuscript nicely written. We also thank the reviewer for the comments and suggestions that have helped us improve our manuscript. Below we provide detailed responses to the comments of the reviewer, and in particular address the major issues raised by the reviewer (comments 1 and 5).

General comment 2: Light matter strong coupling and thus the polaritons are getting immense attention in different fields of science. The work would have significant impact towards the understanding of polaritonic dynamics, if the approximations are little more realistic. Following are the scientific questions / comments.

General Response 2: The reviewer is concerned that the approximations, which were necessary to keep the computations tractable, may affect the impact of our work. We explain below why, despite a significant, but systematic overestimation of the excitation energy at the level of *ab initio* theory employed in our simulations, we believe that our molecular model can capture the underlying mechanisms of the most important processes involved in the propagation of organic polaritons. We thus consider the qualitative insights into the effects of the molecular vibrations on polariton transport, and why these vibrations render the propagation process diffusive rather than ballistic, valid.

Comment 1: The authors previously also have reported various interesting studies with Rhodamine molecules inside an optical cavity. I have a strong concern about the basic computation characterization of the molecule. Experimentally Rhodamine molecules have S0 to S1 transition at around 500 nm (~2.5 eV). But in the present manuscripts as well as in the earlier manuscripts by the authors (J. Chem. Phys. 154, 104112, and J. Chem. Theory Comput. 13, 4324), the S0 to S1 transition in Rhodamine is at 4.18 eV. Why there is a huge difference in the energy values? Can it be improved?

Response 1: The reviewer correctly points out that at the level of theory that we employed in our molecular dynamics simulations, the (vertical) excitation energy is overestimated when compared to the absorption maximum of “real” rhodamine. There are two main reasons for that: First, to reduce the computational effort, we used a minimal size Rhodamine model (*i.e.*, the

basic building block for all rhodamines), which we even further approximated in our QM/MM simulations by including only the fused ring system into the QM region, keeping the benzene ring, which is almost planar to the fused ring system and hence not active in the $S_0 \rightarrow S_1$ excitation, in the MM region. Second, we used a rather small basis set and *ab initio* methods for ground and excited states that do not account for dynamic electron-electron correlation. The agreement with experiment can therefore be improved if we use (i) a model of a real Rhodamine (e.g., Rhodamine 123, Rhodamine B, Rhodamine-6G, Sulforhodamine 101, etc., all of which can be modeled at the QM/MM level), in combination with (ii) a more accurate (and expensive) level of *ab initio* theory or (time-dependent) density functional theory, which can capture more accurately the dynamic electron-electron correlations that are lacking in the current description. For our rhodamine model, for instance, calculating the excitation energy using the fully-correlated (via second-order perturbation theory) extended multi-configuration quasi-degenerate 2nd order perturbation theory (xMCQDPT2) approach by the late Alexander Granovski [Granovski, *J. Chem. Phys.* **134**, 214113 (2011)], which is currently one of the best methods available for modeling excited states, already yields a vertical excitation energy of ~2.5 eV, in much closer agreement with experiment, but at a very high computational cost. Too high for running simulations with over a thousand molecules with the resources that are currently available.

However, as is quite common in computational chemistry, a perfect quantitative agreement with experiment for a specific observable, such as excitation energy, is not essential for obtaining qualitative insights into a process, here the propagation of polaritons in organic microcavities. The key finding of our work is that molecular vibrations driving population transfer between stationary dark states and propagating bright states, control the propagation mechanism. A qualitatively correct description of such vibrations requires that the topology of the potential energy surface (PES) is qualitative correct as well. To check that, we have compared the curvature of the region of the PES that was sampled during the simulation at various levels of theory, including the fully correlated xMCQDPT2 approach. The figure, shown below, and now included as Figure S3 in the revised SI, suggests that the topology is sufficiently similar at all levels included in our benchmark. This finding is also in line with our observation that the vibrational modes of Rhodamine are not overly dependent on the level of theory. The similarities between the curvatures of the PESs at the various levels of theory furthermore suggests that the gradients, which drive the molecular dynamics in the ground and excited states are also sufficiently similar. We therefore anticipate that our findings are unlikely to change if at some point in the future, we can use a higher level of theory in our simulations.

Thus, the consequences of using a low level of theory seem limited to a significant overestimation of the vertical excitation energy, which we could easily compensate for by adding an offset to the cavity dispersion in our simulations. Because under the rotating-wave approximation (RWA) and within the single-excitation subspace, which is valid under the weak driving conditions that are typically employed in experiments on exciton transport in organic microcavities, the absolute energy scale is irrelevant, such offset has no direct impact on the dynamics of the molecule-cavity system. Furthermore, as the Rabi splitting depends only on the number of molecules and their transition dipole moment, which is not very sensitive to the level of theory either (Figure S3b, shown below), the offset also does not affect the energy gaps

between the dark and bright states. We therefore consider the level of theory employed in our model an acceptable compromise for investigating the effect of the molecular degrees of freedom on exciton transport in organic micro-cavities by means of molecular dynamics simulations on the resources available today.

We have added a section to the SI, section 2.1.1 “Comparison of CIS/3-21G to higher levels of theory”, to discuss the accuracy of the model in relation to higher levels of theory. In addition, we included a small section in the main text to discuss the impact of using a lower level of *ab initio* theory on our simulations and refer the reader to the validation benchmarks in the SI.

Page 3: “At this level of theory, the excitation energy of Rhodamine is 4.18 eV, which is significantly overestimated with respect to experiments. This discrepancy is due to the limited size of the basis set and the neglect of electron-electron correlation in the chosen *ab initio* method. While including electron-electron correlation into the description of the QM region improves the vertical excitation energy, we show in the Supporting Information (SI) that this does not significantly change the topology of the relevant potential energy surfaces, which determines the molecular dynamics (Figure S3).”

Figure S3: Panel a: potential energies in the electronic ground (S_0 , continuous lines) and excited state (S_1 , dashed lines) as a function of time in an excited-state MD trajectory of a single Rhodamine at the CIS/3-21G level or theory (green), re-evaluated at the xMCQDPT2//SA2-CASSCF(12,12)/cc-pVDZ (black), SA2-CASSCF(12,12)/cc-pVDZ (red), TD-CAM-B3LYP/6-31G(d) (blue) levels of theory. Panel b: the transition dipole moment at the various levels of theory. Note that because the xMCQDPT2 method only corrects the energy, but not the wave function, the transition dipole moment is the same as that of the underlying CASSCF reference. Panel c: the energy gaps between the excited and ground state.

Comment 2: The calculation in a lossy cavity is performed by considering the rate of cavity loss as 66.7 ps^{-1} (cavity photon life time $\sim 15 \text{ fs}$). From where authors arrived at this value? In Page number 4, Column 1, Line 186, Reference 58 is cited. In that Ref., the photon lifetime in a lossy cavity is given as 25 fs.

Response 2: The reviewer wants to know how we determined the rate of cavity loss in our simulations. We arrived at that value for the decay rate based on an estimate of lifetimes representative for Fabry-Perot microcavities with metallic mirrors. Specifically, the value was chosen to be in the ballpark of the cavity mode lifetime of the cavities fabricated by Schwartz *et al.* [*ChemPhysChem* **14**, 125 (2013), originally reference 58, now 66], which were tuned at 588 nm and had a Q-factor of 30. If we made no mistake, we arrived at a lifetime of 10 fs, slightly below the lifetime in our simulations.

We realize, however, that our sentence may suggest that we chose the cavity mode lifetime in our simulations based specifically on the cavities used for the experiments in that paper, whereas we only included that reference to illustrate that the 15 fs we employ is in range of the lifetimes of typical metallic cavities used in experiment. In a perspective on time-resolved spectroscopy of organic cavity polaritons by Thomas Ebbesen [George *et al.* *Faraday Discuss.* **178**, 281 (2015)], the authors report that the cavity lifetimes in their experiments range from 2 - 14 fs. And more recently, Wu *et al.* [*Nat. Comm.* **13**, 6864 (2022)], used a metallic cavity with a 15~fs lifetime for their experiments on energy transfer within light harvesting complex 2 under strong light-matter coupling. Thus, we chose 15 fs as an upper bound for what is experimentally feasible for cavities with metal mirrors. To make this clear we rephrased the sentence:

On Page 4: “.. we also investigated the effect of the cavity mode lifetime on the transport by performing simulations in an ideal lossless cavity with no photon decay (i.e., $\gamma_{\text{cav}} = 0 \text{ ps}^{-1}$), and a lossy cavity with decay rate of 66.7 ps^{-1} . This decay rate corresponds to a lifetime of 15 fs which is in the same order of magnitude as the 2 - 15 fs lifetimes reported for metallic Fabry-Pérot cavities in experiments [40, 66-68].”

Comment 3: Reported Q factor in most of the experiments is around 100. Can it be considered in the calculation, instead of cavity with infinity Q factor?

Response 3: Because experiments reported Q-factors of around 100, presumably cavities with Distributed Bragg Reflector mirrors, the reviewer wonders whether we could consider such Q-factor, instead of an infinite Q-factor, i.e., a cavity with perfect mirrors. Although a lifetime of 15 fs in combination with a photon energy of 3.81 eV at normal-incidence would correspond to a Q-factor of 87, which is quite close to the Q-factor of 100 proposed by the reviewer, we emphasize that claiming such high Q-factor would be artificial, because of the energy-offset we had to add to the cavity dispersion for compensating the systematic overestimation of the S_0 - S_1 energy gap at the level of QM/MM theory used in our work (See Response 1). Nevertheless, we added the Q-factor to the SI, with a small explanation that the actual value is overestimated due to the offset, which as we argued above, has no direct impact on the dynamics of the molecule-cavity systems within the approximations of our model (i.e., RWA and single-photon subspace).

On Page 14 of SI: “With a lifetime of 15 fs and a resonance at 3.81 eV, the Quality-factor, defined as $Q = \omega_{\text{cav}}\tau_{\text{cav}}$, for the lossy cavity would be 87. We note, however, that this Q-factor is artificially high, because to compensate for the overestimated excitation energies at the CIS/3-21G level of *ab initio* theory (section 2.1.1), we added an offset to the cavity dispersion. Under the rotation wave approximation (RWA) and within the single-excitation subspace, valid under the weak driving conditions that are typically employed in experiments on exciton transport in organic micro-cavities, the absolute energy scale is irrelevant and therefore adding such offset has no direct impact on the dynamics of the molecule-cavity system.”

The answer to the question whether we could consider in the calculations a Q-factor of 100, or any other value, instead of infinity, is yes, we can. The reason for using an infinite Q-factor was to investigate polariton transport without radiative decay, *i.e.*, in a perfect cavity. Because radiative decay from polaritonic states competes with population exchange into the dark states, simulations with “infinite-Q” factor cavities were essential to isolate the contribution of these population exchanges and understand their role in the overall polariton transport mechanism. After we had established that the diffusive transport mechanism at longer timescales is due to such reversible population transfers between the polaritonic and dark states, we made our simulations more realistic by introducing cavity losses, employing a cavity mode lifetime that is in range of the lifetimes in metallic micro-cavities that have been used in experiments. Because we were interested in the effects of cavity loss in general, we did not systematically vary the Q-factor in the current manuscript.

Furthermore, experimentally, the effect of the Q-factor has so far only been investigated for polariton transport initiated by exciting a wavepacket composed mostly of upper polariton states [Pandya *et al.* [*Adv. Sci.* **9**, 2105569 (2022)]]. In that paper, the authors report the results of femtosecond transient absorption microscopy (fs-TAM) on cavities with different Q-factors and conclude that the propagation velocity depends on the Q-factor, for which, according to them, there is no theoretical explanation. To resolve that controversy, we performed new simulations, starting with a wavepacket of UP states, in which we systematically varied the Q-factor. The results of those simulations not only reproduce the experimental observations, but also provide an explanation for the results of Pandya *et al.*, based on our established theoretical model. To avoid that this important message is lost on the reader, we have decided to dedicate a separate publication to that work, available as a preprint on Arxiv [ArXiv:2304.13123, ref. 81], and keep the current manuscript focussed on the general aspects of the mechanism by which polaritons propagate in organic micro-cavities.

Comment 4: What is the Q factor of the lossy cavity?

Response 4: The reviewer wants to know the Q-factor of the lossy cavity. As we mentioned in the previous response, that Q-factor would be 87, but that number is artificially high because of the offset we had to add to the cavity dispersion to compensate for the systematic overestimation of the excitation energy at the CIS/3-21G//Amber03 level of theory employed in our simulations. For a cavity with a lifetime of 15 fs and tuned at the more realistic rhodamine absorption wavelength around 500 nm, the Q-factor would be 53.3 instead.

Whereas the Q-factor is affected by the systematic offset, the cavity lifetime (15 fs, corresponding to a decay rate of 66.7 ps^{-1}) is not. The lifetime also is the physically relevant parameter in our simulations. We therefore prefer to not refer to the Q-factor in the main text, but rather mention it only in the SI, with a small addition to point the reader to the fact that the Q-factor appears higher due to the offset added to the cavity dispersion (see also response 3, above). In the main text we restrict ourselves to mentioning only the cavity mode lifetime.

On Page 14 of SI: “With a lifetime of 15 fs and a resonance at 3.81 eV, the Quality-factor, defined as $Q = \omega_{\text{cav}} T_{\text{cav}}$, for the lossy cavity would be 87. We note, however, that this Q-factor is artificially high, because to compensate for the overestimated excitation energies at the CIS/3-21G level of *ab initio* theory (section 2.1.1), we added an offset to the cavity dispersion.”

Comment 5: In the lossless cavity, the cavity loss is absent, however what about the loss from the molecular decay? In the on-resonant conditions, after 100 fs, excitonic population is increasing. So, if molecular decay is considered, it should generate GS of the molecule. It would be nice to have some thoughts on it.

Response 5: The reviewer wants to know if molecular deactivation, presumably via internal conversion at or near the S_1/S_0 conical intersection seam between the PESs of the electronic ground and excited states [Yarkony, *Chem. Rev.* **112**, 481 (2012)], has been considered as a decay channel when cavity losses are absent. Because without cavity losses, the lifetime of the excited molecule-cavity system will be limited by the excited state lifetime of the molecules, this is a relevant question.

In our QM/MM simulations, the excited state lifetime of typical fluorophores (ns) is much longer than the timescale we can reach (few ps at most). To confirm this for our Rhodamine model, we have located the minimum energy S_1/S_0 conical intersection (MECI), where the non-adiabatic coupling between the electronic excited and ground states is strong and deactivation can occur [Boeije and Olivucci, *Chem. Soc. Rev.* **52**, 2643 (2023)]. Because the MECI is 1.3 eV above the Franck-Condon region of the S_1 PES, we consider it highly unlikely that on the timescale of our simulations, a Rhodamine would reach the conical intersection seam and undergo a radiationless internal conversion back to the ground state.

Therefore, the radiationless decay through internal conversion can be neglected in our simulations, even for cavities with an infinite lifetime. If at some point in the (near) future, longer simulation times will become accessible, the radiationless internal conversion channels can be included explicitly in our simulations by adding the zero-photon ground state in combination with the molecular non-adiabatic couplings, which can be computed analytically in most quantum chemistry codes, to the multi-scale Tavis Cummings Hamiltonian.

If sufficiently long simulation timescales become accessible, we expect that radiationless decay at or near the S_1/S_0 conical intersection seam, where the non-adiabatic coupling between the electronic ground and excited states is large, will set the upper limit to exciton propagation in organic micro-cavities. Likewise, we speculate that also in experiments with an extremely high-Q cavity system with cavity mode lifetimes that are much longer than the typical

nano-second excited state lifetime of the dye molecules (or infinite, like in our simulations), the excited-state lifetime of the molecules would ultimately determine the upper limit of the polariton-mediated energy transport, rather than the cavity lifetime. However, such cavities have so far not been used to investigate exciton transport in the strong coupling regime.

In the main text, we now mention that internal conversion channels are not included in our simulations, and motivate why:

On page 4: “In addition to cavity loss, also internal conversion via the conical intersection seam between the S_1 and S_0 potential energy surfaces[69], can provide a decay channel for the excitation. However, because in our Rhodamine model, the minimum energy conical intersection is 1.3 eV higher in energy than the vertical excitation (SI), and is therefore unlikely to be reached on the timescale of our simulations, we neglect internal conversion processes altogether.”

Comment 6: Polariton transport dynamics is reported by exciting the lower polariton (LP) and dark states (DS) in on resonant and off-resonant conditions respectively. When there is an efficient population transfer between the LP and DS, what is role of excitation energy?

Response 6: The reviewer asks what is the role of the excitation energy when there is population transfer between the lower polariton states and the dark states.

Because the propagation is a diffusion process from the start after an off-resonant excitation of the molecule-cavity system, and eventually becomes a diffusion process after an on-resonant excitation as well, one may argue that the initial excitation conditions do not really matter on longer timescales. However, even if at longer timescales, the mechanism of propagation is the same for both excitation conditions, the initial velocity in the ballistic regime can be tuned via the excitation energy, as the excitation energy (and incident angle of the beam with respect to the cavity normal) determines which superposition of states are excited on the LP branch. Varying the excitation energy in the on-resonant excitation of a wave packet of LP states will thus change the center of that wavepacket. This not only alters the central group velocity of the wavepacket, and hence the propagation velocity in the initial ballistic phase of the transport, but also the energy gap between the photo-excited LP states and the dark state manifold. Because the non-adiabatic coupling for population transfer between the LP and dark states is inversely proportional to that energy gap [Tichauer *et al. J. Phys. Chem. Lett.* **13**, 6259 (2022)], the excitation energy with which the initial wavepacket is excited also affects the efficiency of the population transfer during the ballistic propagation. We have not systematically investigated this effect because doing so would require us to repeat the simulations for multiple initial excitation energies, which is computationally prohibitive at the moment, but speculate on the role of the excitation energy.

On page 7: “Furthermore, because the rate of population transfer is inversely proportional to the energy gap [73], and hence highest when the LP and dark states overlap [71], we speculate that the turn-over line between the ballistic and diffusion regimes depends on the overlap between the absorption line width of the molecules and the polaritonic branches, and can hence be

controlled by tuning the excitation energy to move the center of the initial polaritonic wavepacket along the LP branch.”

Furthermore, because only bright states have in-plane momentum, the direction of the transport during the ballistic phase in a bi-directional cavity can be precisely controlled for on-resonant excitation. In contrast, because dark states lack in-plane momentum and population can transfer into all bright states, irrespective of their in-plane momentum, off resonant excitation into a dark state would lead to propagation in *all* directions. We also highlight this difference in our revision.

On page 7: “In addition, the direction of ballistic propagation can be controlled by varying the incidence angle of the on-resonant excitation pulse.”

And on page 10-11: “Because dark states lack in-plane momentum, the reversible population exchange between dark and bright states causes diffusion in *all* directions. Therefore, under off-resonant excitation conditions, the propagation direction cannot be controlled. In contrast, because bright states carry momentum, the propagation direction in the ballistic phase can be controlled precisely by tuning the incidence angle and excitation wavelength under on-resonant excitation conditions.”

Comment 7: In continuation with the comment 5, especially in the lossy cavity, why the dynamics is different upon on and off resonant excitation?

Response 7: The reviewer asks us to clarify why the dynamics of the wavepacket is different between off- and on-resonant excitation, as well as why this difference is more pronounced in the lossy cavity system.

Immediately after on-resonant excitation of a wavepacket of LP states, the propagation of this wavepacket is ballistic such that it moves with its central group velocity while broadening due to the range of group velocities of the bright states that form the wavepacket. In the absence of cavity losses, only population transfers from these bright states into dark states compete with the ballistic propagation. Because dark states lack group velocity, any population transferred into dark states stops propagating, slowing down the overall wavepacket. However, because population transfers are reversible, population exchanges continuously between LP states *with* group velocity and dark states *without* group velocity. Because the dark states lack in-plane momentum, transfer into the bright states on the LP (or UP) branch can occur at any wave vector. Due to this reversible exchange of population between stationary states and bright states within a range of group velocities, the initial ballistic propagation becomes a diffusion process.

In lossy cavities, the propagation mechanism is the same except that, in addition to the non-adiabatic population transfer into the dark state manifold, bright states also lose population via radiative decay through the imperfect cavity mirrors. This competing radiative loss channel therefore sets an upper limit to the distance over which organic polaritons travel.

If we start the simulation with an off-resonant excitation into the S_1 state of a single molecule, the initial state is not an eigenstate of the molecule-cavity system, but rather a superposition of all states. Reversible exchange of population between stationary dark states and bright states

with group velocity due to both displacement along vibrational modes parallel to the non-adiabatic coupling vectors [Tichauer *et al. J. Phys. Chem. Lett.* **13**, 6259 (2022)] and Rabi oscillations, leads to a continuous formation of wavepackets within the full range of bright states that propagate ballistically until the population transfers back into dark states. Therefore, in contrast to on-resonant excitation, polariton propagation appears as a diffusion process, already from the beginning. Radiative decay in lossy cavities selectively depletes the propagating bright states and hence limits the propagation distance.

Thus, while at *longer times* (*i.e.*, above 100 fs in our simulations), polariton transport is a diffusion process after both resonant and non-resonant excitation, at *shorter time scales*, there is ballistic propagation only under on-resonant excitation conditions. Because radiative decay affects bright states, but not dark states, cavity losses are more prominent in the ballistic regime than in the diffusion regime.

Since the difference in propagation between the two excitation schemes is one of our main findings, we want to avoid that this message gets lost on the readers. We are therefore thankful to the reviewer to remind us to reinforce this message. We now re-emphasize the differences between on- and off-resonant excitation and the effect of cavity loss in the conclusions.

On page 10-11: “While for off-resonant excitation of the molecule-cavity system, these exchanges are essential to transfer population from the initially excited molecule into the bright polaritonic branches and start the propagation process, the exchanges limit the duration of the initial ballistic phase for on-resonant excitation. As radiative decay of the cavity modes selectively depletes the population in bright states, ballistic propagation is restricted even further if the cavity is lossy.”

Comment 8: If I understand correct, dark states are superpositions of collective molecular excitations. Can authors comment on the ratio between the polaritonic states to Dark states in their calculation?

Response 8: The reviewer has understood correctly that also in the dark states, the excitation can be shared between multiple molecules, and can hence be considered superpositions of collective molecular excitations. The reviewer suggests that we comment on the ratio between the polaritonic states, which are bright superpositions of molecular excitations *and* cavity mode excitations on the one hand, and the dark states, which are superpositions of molecular excitations only, on the other hand. We agree with the reviewer that including these ratios is important for a reader to better understand the effect of the system size on the observed dynamics since the ratio between the number of cavity modes and molecules must be small enough to have a significant dark states density. We note that a clear distinction between the polaritonic and dark states is possible only if there is no disorder and the molecules are identical. In the presence of disorder, the cavity modes become “smeared” out over many more states with the brighter states with higher cavity mode contributions, still forming the upper and lower polariton branches.

Because the cavity dispersion was modeled with 160 discrete modes in our simulations, the total number of superposition states that can form when N_{mol} molecular excitations are strongly coupled to these modes, is $N_{\text{mol}}+160$. In the ideal scenario of a system without disorder, 320 of these states are polaritonic, with 160 forming the lower and 160 the upper polariton branches, while $N_{\text{mol}}-160$ are dark. Thus, in this ideal case, the ratio between the polaritonic and dark states in our simulations are 0.37 for 1024 molecules, 0.91 for 512 molecules and 3.33 for 256 molecules. We have included these ratios (the inverse, actually) in the SI where we present the details of the systems with different numbers of molecules.

On page 15 in SI: “For systems without disorder, bright states and dark states are easily distinguishable because dark states lack contributions from cavity mode excitations. In contrast, in the presence of disorder due to molecules adopting different conformations in the simulations (and in reality), the cavity modes are smeared out over many states. Nevertheless, because the cavity modes are not distributed evenly over all states, the upper (UP) and lower (LP) polaritonic branches remain visible, even if the number of eigenstates that make up these branches exceeds the number of cavity modes. Therefore, to have sufficient dark states, loosely defined as states with a negligible cavity mode contribution, the ratio between the number of dark (N_{dark}) and polaritonic states (N_{pol}) must be larger than zero. While this is the case when there are 1024 ($N_{\text{dark}}/N_{\text{pol}} = 2.7$), 512 ($N_{\text{dark}}/N_{\text{pol}} = 1.1$), or 256 molecules ($N_{\text{dark}}/N_{\text{pol}} = 0.3$) in the cavity, the smallest cavity systems with 160 molecules ($N_{\text{dark}}/N_{\text{pol}} = 0$), has no real dark state manifold, which, as we show below, affects the propagation mechanism.”

Comment 9: Upon off-resonant excitation to S1 state of the molecule, how the upper polariton is observed (Figure 4c)?

Response 9: The reviewer asks how the UP is observed when we excite a single molecule into the S_1 electronic state. We are not fully sure if we have understood what the reviewer refers to. If it is about the *observation* of the UP in the 2D spectrum of Figure 1, we note that these spectra are based on the eigenstates of the Tavis Cummings Hamiltonian. Our 2D spectra show the “visibility”, a concept introduced by Lidzey *et al.* [Lidzey *et al.* Science 288, 1620 (2000)], and defined as the total photonic weight of a state. These details are now included in a new section 3.4 “Photo-Absorption Spectra” of the SI on page 22.

However, we also considered the possibility that the reviewer wants to know how the *population* enters the UP state if initially a single molecule is excited into the S_1 state by an off-resonant excitation. Such an initial state is *not* an eigenstate of the molecule-cavity Hamiltonian, but rather a superposition of *all* eigenstates. Therefore, population exchange is due to both Rabi oscillations and non-adiabatic population transfer, with the Rabi oscillations dominating this process at the start of the simulation (see also response 7 above).

To avoid confusion about what causes the UP (and LP) states to become populated when off-resonant excitation conditions are applied, and clarify this important aspect of such excitation conditions, we now emphasize this more clearly in the main text:

On page 8: “Since the initial state, with one molecule excited, is not an eigenstate of the molecule-cavity system, population exchange from this state into the propagating bright states is not only due to displacements along vibrational modes that are overlapping with the non-adiabatic coupling vector [73], but also due to Rabi oscillations, in particular at the start of the simulation.”

Comment 10: Additional minor suggestion: In Page 2; Column 1, Line 92, It is written like “The majority of hybrid states in realistic molecule- cavity systems are dark, [20–22] meaning that they have no contributions from the cavity modes,” I think, dark states will have minor contribution from the cavity modes. We can not say that there is no contribution.

Response 10: We agree with the reviewer that a strict distinction between bright and dark hybrid states can only be made if there is no disorder (see also our response to comment 8, above). Since we have disorder in our simulations, all states acquire a minor contribution from the cavity modes, as the reviewer pointed out. We therefore followed the reviewer’s suggestion and changed our statement to:

On Page 1: “The majority of hybrid states in realistic molecule-cavity systems are dark [21-23], meaning that they have negligible contributions from the cavity modes.

Comment 11: The figure legends, molecular and GS are confusing in Fig. 2d, h and 3d, h. The term molecular could be changed to excitonic to have better clarity.

Response 11: We thank the reviewer for pointing out that the labeling in the figure legends was confusing. We therefore followed the suggestion of the reviewer and changed the term “molecular” to “excitonic”, and explicitly wrote “ground state” instead of GS.

Reviewer 2

General Comment 1: The manuscript presents theoretical investigations into polariton-assisted exciton transport and is both timely and interesting. However, there are a few points that could be improved.

General Response 1: We thank the reviewer for the time and effort spent on evaluating our manuscript. We are pleased that the reviewer considers our investigation timely and interesting. We also thank the reviewer for raising important points that have helped us improve our manuscript and reinforce our conclusions.

Comment 1: Firstly, it would be helpful to clarify the role of the Stokes shift in resonant LP excitation (are conclusions specific to Rhodamines?).

Response 1: The reviewer considers it helpful if we clarify the role of the Stokes shift in the resonant excitation of a wavepacket of lower polariton (LP) states. In addition, the reviewer wants to know if our conclusions are specific to Rhodamine.

Our main finding is that on longer timescales, polaritons in organic micro-cavities do not propagate ballistically at their respective group velocities, but much slower and rather in a diffusive manner because of reversible population exchange between polaritonic states with group velocity and dark states without group velocity. The rate of these transitions is determined by the strength of the non-adiabatic coupling vector, which depends on (i) the energy gap between the polaritonic and dark states; and (ii) the difference between the gradients of the ground and excited state electronic potential energy surface (PES) [Tichauer *et al. J. Phys. Chem. Lett.* **13**, 6259 (2022)]. Because the magnitude of the “gradient difference” depends on the displacement of the Frank-Condon active modes (*i.e.*, the Huang-Rhys factor) in combination with their vibrational frequencies [De Jong *et al. Phys. Chem. Chem. Phys.* **17**, 16959 (2015)], the magnitude and direction of the non-adiabatic coupling vector are directly correlated with the Stokes shift.

Since in organic micro-cavities the absorption linewidth of organic molecules, and hence the distribution of dark states (with exception perhaps of J-aggregates), is often broader than the Rabi splitting, the turn-over between coherent ballistic propagation into diffusion for *on-resonant excitation* is mostly determined by the overlap between dark states and states of the LP branch that compose the initial wavepacket (*i.e.*, when the energy gap is zero [Groenhof *et al., J. Phys. Chem. Lett.* **10**, 5476 (2019)]). In contrast, if the Rabi splitting exceeds the linewidth of the molecules, as for J-aggregates, the rate of population exchange would be mostly determined by the molecular Stokes shift, as it controls the magnitude and direction of the non-adiabatic coupling vector [Tichauer *et al. J. Phys. Chem. Lett.* **13**, 6259 (2022)].

Upon *off-resonant excitation*, a single molecule is excited, which subsequently relaxes into the S_1 minimum, from where Stokes-shifted emission normally occurs. Tuning the Rabi splitting such that the region of the LP branch with the highest group velocity matches the energy of the S_1 minimum, could therefore maximize the population transfer into those LP states and enhance

transport, at least initially. Thus, unless the Rabi splitting is larger than the absorption linewidth of the molecules, which for typical organic materials would require entering the regime of *ultra-strong coupling*, we expect the role of the Stokes shift to be more important for transport under off-resonant excitation than under on-resonant excitation. Because the Stokes shift can be an important “control knob” for polariton propagation, we have added a sentence to the conclusions of our manuscript to reflect these insights.

On page 11: “The rate at which population transfers between bright and dark states depends on the non-adiabatic coupling vector, whose direction and magnitude are determined by the Huang-Rhys factor in combination with the frequency of the Franck-Condon active vibrations [73], both of which are related to the molecular Stokes shift [76]. In addition, because the non-adiabatic coupling is inversely proportional to the energy gap [73], the Stokes shift in combination with the Rabi splitting, also determines the region on the LP branch into which population transfers after off-resonant excitation of a single molecule [78-81]. We therefore speculate that the Stokes shift can be an important “control knob” for tuning the coherent propagation of polaritons.”

The reviewer also asks if our conclusions about the polariton propagation mechanism are specific to Rhodamine. With the aim of elucidating the general mechanism of polariton propagation in organic micro-cavities, we had focussed on Rhodamine as a model system. In spite of a systematic overestimation of the vertical excitation energy due to the necessary compromises we had to make in the *ab initio* description of the electronic structure, this rhodamine model features the key photophysical properties of a real organic molecule: vibrational modes, of which few are Frank-Condon active, absorption linewidth (100 meV FWHM), Stokes shift (120 meV). We therefore consider this molecule representative for organic dyes and that the transport mechanism we observed in our simulations is general for polariton propagation in organic microcavities.

To reinforce our conclusions, we have included results from an ongoing project aimed at investigating the effect of molecular properties on propagation. In this ongoing project, we are performing simulations of organic molecules with different Stokes shifts and absorption linewidths, including Methylene Blue, Rhodamine and Tetracene. Because the new simulations are aimed at understanding how specific molecular (and cavity) parameters can be tuned to optimize exciton transport in the strong coupling regime, we consider them beyond the scope of the current manuscript, in which we aim to share our insights into the general mechanism of such transport, which we consider essential to understand the experimental observations, at least qualitatively. Adding the results of simulations that address more specific research questions, would not only delay sharing these insights with the community, but could potentially even distract a reader from our main message.

Nevertheless, to support our conclusion that irrespective of the initial excitation conditions, polariton transport becomes diffusive on longer timescales due to reversible population transfer between propagating polaritonic states and stationary dark states, we have added results of simulations for Methylene blue, with a narrower absorption linewidth than Rhodamine, and for Tetracene with a broader linewidth, to the Supporting Information (SI), section 4.9, “Polariton

propagation in Tetracene and Methylene Blue cavities”, on page 48. The results of these simulations confirm that upon resonant excitation, propagation is ballistic until the excitonic contributions dominate the wavepacket and render propagation diffusive. Because for Tetracene the absorption linewidth (and hence also the distribution of dark states [Groenhof *et al.*, *J. Phys. Chem. Lett.* 10, 5476 (2019)]), and Stokes shift are larger than for Rhodamine and Methylene Blue, the rate of population transfer from the initially excited LP states into the dark state manifold is also higher. As a consequence, the cross-over between the ballistic regime and diffusion regime occurs almost immediately for tetracene. Nevertheless, despite these differences, which are due to variations in the photophysical properties of molecules, the overall mechanism remains identical.

To emphasize why we consider Rhodamine representative of organic molecules we have added the following sentences to the Conclusions, which we further support by referring to the additional simulations of Tetracene and Methylene Blue:

On page 11: “Because our Rhodamine model features the key photophysical characteristics of an organic dye molecule, we speculate that the propagation mechanism observed in our simulations is generally valid for exciton transport in strongly-coupled organic micro-cavities, in which the absorption line-width of the material exceeds the Rabi splitting and there is a significant overlap between bright and dark states. To confirm this, we have also performed simulations of exciton transport in cavities containing Tetracene and Methylene Blue and observed that the propagation mechanism remains the same (SI).”

Comment 2: Secondly, I am curious to know if the authors have any comments on the guided exciton-polaritons that exist below the light line, and their role in enhanced exciton transport (it appears that the authors have only discussed exciton coupling to FP modes above the light line).

Response 2: The reviewer asks us to comment on the guided exciton polaritons that exist below the light-line and their role in enhancing exciton transport. The reviewer correctly points out that in our work, we have only considered transport when excitons are coupled to the modes of a metallic Farby-Pérot cavity. In addition to coupling to the Farby-Pérot modes in such cavities, molecules can also couple to surface plasmon polaritons (SPP) that are supported by the metal surfaces constituting the mirrors and that exist below the light line. While their role will depend on the details of the setup (*e.g.*, the materials used, energy of the relevant molecular excitations, *etc.*), we can provide some general observations on their expected influence. As the SPPs decay exponentially away from the surface, they will in particular affect molecules close to the surfaces, while standing-wave Farby-Pérot modes (in particular the lowest-order one) preferentially couple to molecules in the center of the cavity due to the sine profile of the mode. The spatial overlap between the modes (and thus molecules) accessed by driving the cavity within the light line and the SPP modes is thus not expected to be very large. Still, once the polaritons have transferred energy to the reservoir modes / dark states, the interchange of energy between the reservoir and the polariton modes can *a priori* also happen with the polaritons obtained from hybridizing molecular excitons and SPP modes, adding an additional transport channel (as SPPs and exciton-polaritons formed by them generally have high group

velocities) with general characteristics comparable to the analogous process we observe with the FP-cavity modes. While this extra channel is not directly visible in experiments measuring emission within the light line, the energy transfer from the reservoir to SPP-exciton polaritons and back to the reservoir could contribute to the effective diffusion constant that is observed in experiment, although the qualitative behavior is not expected to change.

We have added a brief statement to comment on the involvement of guided-exciton-polaritons that exist below the light line.

On page 10: “In our simulations we couple excitons only to the modes of the Fabry-Pérot cavity, whereas in experiments with micro-cavities constituted by metal mirrors, excitons can in principle also couple to surface plasmon polaritons (SPP) below the light line that are supported by these metal surfaces. While their role will depend on the details of the set-up (e.g., the materials used, energy of the relevant molecular excitations, etc.), we cannot rule out that reversible population transfer between the dark states and SPP-exciton polaritons also contributes to the effective diffusion constant observed in those experiments [40,48]. However, because the SPP decays exponentially away from the metal surface, and SPP-exciton polaritons also have group velocity, the qualitative behavior is not expected to change.”

Comment 3: Furthermore, with regards to LP->DS transfer, it would be valuable to explore the role of the Stokes shift, Rabi splitting and temperature in this process. It seems to me that it could have been beneficial to consider working with zero Stokes shift systems like J-aggregates, while maximizing the Rabi splitting.

Response 3: While we fully agree with the reviewer that exploring the roles of the Stokes shift, the Rabi splitting and temperature are important directions for further research, we consider extending the current investigation into these directions beyond the scope of the present manuscript, which we want to keep focussed on the general mechanism by which polaritons can propagate in organic microcavities. Because it is important, in particular for applications, to understand and predict how molecular properties, temperature, and cavity parameters affect the transport of excitons, we have already initiated new projects, aimed at addressing precisely these questions.

While molecular properties (such as absorption line-width, or Stokes shift) or temperature are relatively straightforward (although computationally demanding) to alter in our simulations by changing the molecule, or thermostat temperature, exploring the effect of the Rabi splitting is less straightforward. If, as is commonly done in experiments, we change the Rabi splitting (proportional to the square root of the number of molecules) by varying the number of molecules in our simulations, we would alter also the number of dark states and the rates of population transfer (which depend on N as well [Tichauer *et al. J. Phys. Chem. Lett.* **13**, 6259 (2022)]). It therefore will be challenging to disentangle the effect of changing the Rabi splitting from the effect of changing the number of molecules (or dark states) in our simulations.

Alternatively, we can change the Rabi splitting by varying the cavity vacuum field strength (which is inversely proportional to the cavity mode volume), which is difficult in experiments, but

easy in our simulations. However, because changing the field affects the non-adiabatic coupling vectors as well, it will not be straightforward either to attribute changes in transport to Rabi splitting alone in such simulations.

Notwithstanding these challenges, we repeated the simulations for 1024 molecules in a lossy cavity at three different Rabi splittings by varying the vacuum field strength for both on- and off-resonant excitation. These results, which are included in the SI, subsection 4.7 “Effect of Rabi splitting on polariton propagation” on page 44, do not suggest a clear trend, except that the initial propagation velocity in the ballistic phase immediately after on-resonant excitation follows the decrease of the maximum LP group velocity with increasing Rabi splitting. Further simulations are therefore needed (and currently undertaken) to correlate the Rabi splitting to the transport properties. While we have developed a concept to vary the Rabi splitting in our simulations without changing N or the vacuum field, we will need more time to work this out. Since we consider it unlikely that these simulations will change our view on the general mechanism of polariton propagation in organic materials, which is the main message we wish to convey in the present manuscript, we intend to share those results in a future publication. In the current manuscript, we have added a sentence to alert the reader that work is in progress on investigating the effects of temperature, Rabi splitting and molecular properties on polariton transport:

On page 11: “Future work will be aimed at investigating how the propagation can be controlled by tuning molecular parameters, temperature, Rabi splitting, or cavity Q-factor [82].”

The reviewer suggests that it could have been beneficial to work with a zero-Stokes shift system in combination with a maximum Rabi splitting. We agree that a zero-Stokes shift system, in particular if combined with a narrow absorption linewidth, such that the polariton branches do not overlap energetically with the dark states [Groenhof *et al.*, *J. Phys. Chem. Lett.* 10, 5476 (2019)], can provide further insights into the mechanism by providing an “extreme” case that is different from typical organic molecules (except for J-aggregates) with moderate to large linewidths and Stokes shifts. We note, however, that our simulations with frozen nuclear degrees of freedom already captured the most important aspects of such a “zero Stokes shift, zero absorption linewidth” system, and that the results from these simulations indeed confirm that the propagation remains coherent and hence fully ballistic during the polariton lifetime, as we explain in further detail below.

Under the common assumption that the Stokes shift is dominated by the displacement rather than by changes of the mode frequencies in the electronic excited state [De Jong *et al. Phys. Chem. Chem. Phys.* 17, 16959 (2015)], the non-adiabatic coupling vectors for transitions between the bright and dark states would be very small, as there would be no difference between the S_1 and S_0 gradients for a system with a “zero Stokes shift” [Tichauer *et al. J. Phys. Chem. Lett.* 13, 6259 (2022)]. Instead, the non-adiabatic coupling vector would be dominated by derivatives of the inner-product between the vacuum field strength and the transition dipole moment with respect to nuclear motions. Because for realistic cavity fields, this term would be very small, we consider that the frozen Rhodamine system with all nuclear degrees of freedom constrained, is essentially the same as a zero-stokes shift, zero line-width two-level system,

which in turn could be considered a (rough) approximation for a J-aggregate. Simulations of the frozen system suggest that with no non-adiabatic coupling, nor overlap between the bright LP branch and dark states, on-resonant excitation leads to fully coherent ballistic propagation, limited only by the finite lifetime of the cavity modes. We therefore speculate that this is what will happen when J-aggregate polaritons are resonantly excited. In contrast, for off-resonant excitation of a J-aggregate cavity system, we anticipate that the diffusion will be slow due to the small magnitude of the non-adiabatic coupling between the initially populated dark states and the bright polaritonic states. The results of the simulations with the frozen Rhodamine system are presented in the SI, section 4.3, "Simulations at 0 K" on page 37.

To overcome the limitations of the frozen system, we had experimented with adding constraints only on bond lengths during the simulations, which for the conjugated dyes that are typically used in experiments, prevent relaxation of the molecule in the excited state and hence suppress the Stokes shift. With such constraints on the bond lengths only, the molecule still has a narrow absorption linewidth, in contrast to the frozen Rhodamines, and would thus be a better approximation to a J-aggregate. While in these simulations, ballistic propagation dominates, there is some population transfer into the dark state manifold. However, as the energetic overlap is small, as well as the magnitude of the non-adiabatic coupling vector, the rate of this transfer is low and no turnover from ballistic to diffusion could be observed on the timescale of the simulation. Because the results of this system are similar to that of the frozen system, but represent an artificial situation, in which the constraints alone could cause a reduction in population transfer, and could potentially obscure our main message, we have not included these results to the manuscript.

Comment 4: Finally, I believe there appears to be a typo in line 149 which should read "provide".

Response 4: We thank the reviewer for spotting this typo, which is now corrected.

Comment 5: Overall, this manuscript deserves the attention of the scientific community.

Response 5: We thank the reviewer for these encouraging words.

Reviewer 3

General comment: In this manuscript, Sokolovskii performed the multiscale MD simulations of exciton polariton propagation in a 1D Fabry-Perot cavity. Results with two different initial conditions were discussed: (i) exciting the LP mode at the maximal group velocity; (ii) exciting the S₁ state of a single molecule. Their results showed that the LP propagation along the cavity mirror plane is initially ballistic and becomes diffusive in later times. They further analyzed the origin of this diffusive behavior and studied the important roles of the molecular dark modes. As the first atomistic simulation of exciton polariton propagation along the cavity mirror plane, this manuscript is likely to attract a broad audience of Nature Communications. Hence, I would recommend the publication of this manuscript if the following points are properly addressed.

General response:

We thank the reviewer for evaluating our manuscript. We agree with the concise summary and are pleased that the reviewer recognizes that our manuscript reports the first atomistic simulation of polariton propagation in organic microcavities. By addressing the points raised by the reviewer, we could improve the presentation of our work in the revision of our manuscript.

Comment 1: According to the theory of multi-scale MD for exciton-polaritons, only the first excited states of molecules and photons are included. Upon exciting the polariton or S₁ state, this approximation is valid if the time duration is short. In later times (e.g., the diffusive motion of polaritons), however, the total wavefunction of all molecules + photons could contain double excitations (e.g., two molecules in the first excited states), which is not described within this framework. Could the authors estimate the effects of the missing double excitations in their simulations?

Response 1: The reviewer correctly points out that we only included the first excited states of the molecules and single photon Fock states of the cavity modes into our simulations and asks us to estimate the effects of missing double excitations in our simulations.

Because the vacuum field strength is sufficiently weak in our simulations, our system remains in the strong coupling regime, where the rotating wave approximation (RWA) can be considered valid. We thus neglect counter-rotating terms, which couple manifolds that differ by two excitations (one in a molecule and one in a cavity mode), which would require significantly larger Rabi splitting to become important. Adding these counter rotating terms in our simulations also requires a much larger basis that would also include states with zero, two, three, four, etc. excitations. With such an extended basis, the initial state after off-resonant excitation into a single molecule, would thus be

a superposition of all states, including also basis states with double, triple, etc. photon excitations, in which multiple molecules are excited. Therefore, the initial state would no longer correspond to the situation we try to model, namely a single molecule in the excited state, which we consider the most relevant scenario for the experiments.

We also assume that the pump pulse is sufficiently weak for the system to remain in the linear response regime. Thus, upon interacting with the system, the pulse delivers only a single photon. Under these conditions, the restriction to the single-excitation subspace is valid. Furthermore, as long as there are many more molecules than photons in the system, this restriction would remain valid also when multiple photons are absorbed [Feist *et al.*, *ACS Photonics* **5**, 205-216 (2018)]. We therefore do not expect to see major differences in the transport mechanism if we would have included excitation of another molecule.

To make our assumptions about the excitation conditions in our work more clear and avoid confusion, we have added the following:

On page 4: “We assume that the intensity of the excitation pulse in both cases is sufficiently weak for the system to remain within the single-excitation subspace. We thus exclude multi-photon absorption and model the interaction with the pump pulse as an instantaneous absorption of a *single* photon.”

Comment 2: One largest issue on simulating polaritons is the so-called “large N problem”: because it is very hard to explicitly simulate millions (the experimental value) of chromophores, one can only simulate a finite (usually tens to thousands) number of chromophores and then check the system size dependence of the observables by fixing the Rabi splitting. The macroscopic limit, or the large N limit, should correspond to the experiments. The authors did this system size check in the SI, but in the main text it is relatively hard to find the N- dependence of many interesting observations, especially the reversible energy transfer from dark modes to polaritons in the resonant excitation case (Figs. 2 and 3) and the off- resonant excitation case (Fig. 4 and 5). Hence, I strongly suggest the authors to briefly discuss the N-dependence for their observations in the main text: what the results would be if N approaches the experimental limit?

Response 2: The reviewer suggests that we move part of the discussion on the finite size effects, often referred to as the “large N problem”, from the Supporting Information (SI) into the main text. We agree and have now included the following text in the main manuscript. We comment in detail on the extrapolation of our results to larger N in response 4 below.

On page 10: “**Size dependence**

Due to limitations on hard- and software, the number of molecules that we can include in our simulations is much smaller than in experiments [74-76]. We therefore investigated how the number of molecules N coupled to the cavity affects the propagation by performing simulations for different N . To keep the Rabi splitting (~ 325 meV) constant, and hence polariton dispersion the same, we scaled the cavity mode volume by the number of molecules N (see SI for details).

While the transport mechanism is unaffected by N (Figures S4-S12), the total population that resides in the bright states decreases when the number of molecules, and hence the number of dark states, increases, in particular in the diffusion phase. Such decrease in bright state population is due to the $1/N$ scaling of the rate at which population transfers between dark and bright states [73]. Because the number of dark states is proportional to N , while the number of bright states is constant for a fixed number of cavity modes, this dependency affects the ratio between the population in the dark and bright states, with the latter rapidly decreasing with increasing N . As the overall propagation velocity is determined by the population in bright states, also the velocity is inversely proportional to N (Figure S13). Therefore, in experiments, with 10^5 - 10^8 molecules inside the mode volume [74–76], the propagation velocity will be much lower than in our simulations.

Nevertheless, because of the $1/N$ scaling, the effective polariton propagation velocity approaches the lower "experimental limit" of 10^5 coupled molecules [74] already around 1000 molecules. We therefore consider the results of the simulations with 1024 Rhodamines sufficiently representative for experiment and for providing qualitative insights into polariton propagation. Indeed, a propagation of $9.6 \mu\text{mps}^{-1}$ in the cavity containing 1024 molecules is about an order of magnitude below the maximum group velocity of the LP ($68 \mu\text{mps}^{-1}$) in line with experiments on organic microcavities [40], and cavity-free polaritons [43].”

Comment 3: Relating to the above comment, in page 5, the authors stated, “*This re-spawning process leads to the diffusive propagation of the excitation observed in Figure 2, with an increasing wavepacket width (Figure 3a), in line with experimental observations [37, 39, 40, 45].*” I would assume that the re-spawning process would become very weak when N is large, no?

Response 3: The reviewer asks if the assumption that the re-spawning process, in which population transfers from the dark states into the bright polaritonic states, would become weaker when the number of molecules (and hence the number of dark states) increases. That is correct. We confirm that this is what our simulations suggest, and that this is why the observed propagation velocity reduces when N increases. In previous work we could show that the non-adiabatic coupling for population transfer between

bright and dark states is inversely proportional to N [Tichauer *et al. J. Phys. Chem. Lett.* **13**, 6259 (2022)]. Because the density of polaritonic states is fixed by the number of modes, whereas the density of dark states increases linearly when we add more molecules to the cavity, the rate of population transfer from dark states into the polaritonic states reduces, while the rate in the opposite direction remains the same. However, in contrast to the usual definition of the thermodynamic limit, where N would go to infinity, the relevant limit in polaritonic chemistry is the maximum concentration, *i.e.*, the maximum number of molecules, N , inside the finite mode volume of the cavity, V_{cav} . While that number can be large, it cannot be infinite and estimates range from 10^5 [Houdre *et al. Phys. Rev. A*, **53**, 2711 (1996)] to 10^8 [Eizner *et al. Sci Adv.* **5**, eaax4482 (2019)]. In this limit, the non-adiabatic coupling would be smaller than in our simulations, but not zero. Thus, in spite of a reduced rate of population transfer compared to our simulations, the propagation would still occur, albeit with an even smaller diffusion coefficient and longer lifetime, which is in line with experiments on organic microcavities measuring significantly longer lifetimes than in our simulations. To reflect these insights and emphasize one important source for systematic deviation between our simulations and experiments, we have added the following sentence to the manuscript (see also our response 2, above):

On page 10: “As the overall propagation velocity is determined by the population in the polaritonic states, also the velocity is inversely proportional to N (Figure S13). Therefore, in experiments, with 10^5 - 10^8 molecules inside the mode volume [74-76], this rate is much lower than in our simulations.”

Comment 4: In page 7, the authors stated, “Because the overall propagation velocity is determined by the population in the bright states, and that population is inversely proportional to N due to the inverse scaling of the non-adiabatic population transfer rate [62], also the velocity is inversely proportional to N (Figure S12).” In the SI, the authors fitted the N dependent results of the average propagation velocity $v_{\text{av}} = 1/N + 1/v_{N \rightarrow \infty}$ and claimed using 1000 molecules is good enough to get the macroscopic limit. Why do the authors not use the formula $v_{\text{av}} = 1/N$ to do the fitting? Of course, with this new formula one would get zero v_{av} in the macroscopic limit (which is not good to compare with experiments). I would assume that if only a single molecule is initially excited, no polariton excitation would be observed at all in the experimental limit. In experiments, the off-resonant excitation probably excites a finite fraction of molecules instead of a single molecule, so the enhanced diffusion could be observed in experiments.

Response 4: The reviewer asks why we have included an additional constant $1/v_{N \rightarrow \infty}$ in the fits to the polariton propagation velocities as a function of the number of molecules. The reviewer correctly points out that a fit without such constant would lead to a zero velocity in the macroscopic limit, but as we argue in our response above, the relevant

limit for organic polaritons is not where N goes to infinity, but rather to a finite number, which can be high, higher at least than we can handle in our simulations. Nevertheless, we agree with the reviewer that the $1/v_{N \rightarrow \infty}$ constant has no simple physical explanation and that adding this constant may thus not be the most suitable way to perform the fit. Leaving out the constant, as suggested by the reviewer, however, leads to a worse fit, which we attribute to the small molecule-to-cavity-mode ratio for the smaller ensembles. To circumvent the issue that the deviations from inverse N scaling, that are most prominent when N is small, we instead fitted the function $v_{av} = a/(N+b)$ to the velocity as a function of N . The constant b accounts for deviations from the anticipated $1/N$ scaling when the ensembles are small, while it disappears when the ensemble sizes reach the macroscopic limit. With this new fit, we predict a finite propagation velocity even in the macroscopic limit, *i.e.*, $v_{av} = 100$ nm/ps for 10^5 molecules, $v_{av} = 10$ nm/ps for 10^6 molecules and $v_{av} = 1$ nm/ps for 10^7 molecules, all of which significantly exceed the average velocity at which excitons propagate in bare materials.

While we cannot exclude that in experiment multiple molecules are excited, our simulations suggest that exciting a single molecule in an ensemble of collectively coupled molecules can already initiate wavepacket propagation. In addition to updating the SI with the new fit (Page 33-35 and Figure S13), we have added further discussion about the N dependence of the propagation velocity, including a justification for the new fitting procedure, in the main text (see also responses 2 and 3):

On page 10: “Nevertheless, because of the $1/N$ scaling, the effective polariton propagation velocity approaches the lower “experimental limit” of 10^5 coupled molecules [74] already around 1000 molecules. We therefore consider the results of the simulations with 1024 Rhodamines sufficiently representative for experiment and for providing qualitative insights into polariton propagation. Indeed, a propagation of $9.6 \mu\text{mps}^{-1}$ in the cavity containing 1024~molecules is about an order of magnitude below the maximum group velocity of the LP ($68 \mu\text{mps}^{-1}$) in line with experiments on organic microcavities [40], and cavity-free polaritons [43].”

Comment 5: In page 4, the authors stated, “Because no peaks are observed if the molecular degrees of freedom are frozen (Figure S14), we infer that these localizations are a manifestation of energetic disorder among the molecules that affects their contribution to the wavepacket [32].” From my understanding, Fig. S14 seems to indicate that in the absence of thermal motion, polaritons would not dephase to the dark modes, while Fig. 2 appears to show that with thermal motion, polaritons can dephase to the dark modes. Such a dephasing process is of course temperature dependent. I cannot understand why it is related to the localization of polaritons due to energy disorder.

Response 5: The reviewer correctly points out that without nuclear motion, the polaritons do not dephase into the dark states, but asks why such dephasing at elevated temperatures is related to localization of polaritons due to energy disorder, as we had argued in our manuscript. Based on our results, the partial localization of the wavepacket is not necessarily related to the dephasing, as also in simulations at 0 K of a system with energetic disorder among the excitons, such peaks are visible in the ballistic regime (see figure below). We note that in the simulations with frozen nuclear coordinates in Figure S14 (now Figure S15), there was no disorder among the molecules (*i.e.*, the coordinates were identical for all molecules). Also, if we already include disorder at the start of the MD simulations by selecting coordinates from the QM/MM equilibrium trajectory of uncoupled Rhodamine at 300 K, spikes appear immediately (see figure S21, section 4.6 “Effect of initial energy disorder on ballistic transport” in SI). Similar observations were made by Agranovich and Gartstein (ref. 30 in the original submission, ref. 35 in the revision) who attributed these peaks to (partial) localization due to energy disorder among the molecules. We thus follow their interpretation. To make this interpretation more clear in the main text we have added the following text.

On page 5: “Such peaks are not observed if there is no disorder and the molecular degrees of freedom are frozen (Figure S15), but appear already at the start of the simulation when the initial configurations of the molecules are all different (Figure S21). Similar observations were made by Agranovich and Gartstein [35], who attributed these peaks to energetic disorder among the molecular excitons. We therefore also assign these peaks to a partial localization of the wavepacket at the molecules due to structural disorder that alters their contribution to the wavepacket.”

Effect of initial disorder on the wavepacket propagation of 1024 molecules at 0 K. The extent of the disorder is defined as $\sigma = 50$ meV, the FWHM of a Gaussian distribution of excitation energies, from which the initial excitons are selected.

Comment 6: Overall, this manuscript is of high quality, and reading it is a delightful experience.

Response 6: We thank the reviewer for these kind and encouraging words.

REVIEWER COMMENTS

Reviewer #1 (Remarks to the Author):

The revised manuscript by Sokolovskii et. al. entitled, “Enhanced Excitation Energy Transfer under Strong Light-Matter Coupling: Insights from Multi-Scale Molecular Dynamics Simulations” deals with the excitation energy transfer between Rhodamine molecules under electronic strong coupling conditions. The Authors satisfactorily answered and made corresponding corrections in the revised manuscript. However, some more queries arise from the responses made by the authors.

1. The length of the cavity is taken as 163 nm. In such conditions, which mode of the cavity is coupled with the molecular transition? If the fundamental mode of the Fabry-Perot cavity is used, is it matching with the calculated transition energy (4.18 eV) or experimental transition energy (2.5 eV)?
2. In page 10, the Authors newly added a section about size dependence. Here, it is mentioned that the transport mechanism is not strongly affected by the number of molecules. Then what is the role of collective coupling in these observations?
3. The Authors made a rough estimation of the ratio of polaritonic states to Dark states in the response letter and supporting information. The calculated numbers are independent of a molecule under study. It is only about numbers. Is there any role of the strength of strong coupling (Rabi splitting) in determining the ratio mentioned above?
4. Page 10, column 1, the last sentence is unclear.

Reviewer #2 (Remarks to the Author):

In the revised manuscript, the authors have addressed all of my comments and concerns. They have provided satisfactory explanations for cases where they could not fully address my questions, and they have also mentioned that those aspects might be the focus of future work. Based on these revisions, I recommend the revised version for publication in Nature Communications.

Reviewer #3 (Remarks to the Author):

The authors have properly responded to my previous comments and the quality of the paper is beyond expectation. I enthusiastically recommend the publication of this paper.

Reviewer 1

General Comment: The revised manuscript by Sokolovskii et. al. entitled, “Enhanced Excitation Energy Transfer under Strong Light-Matter Coupling: Insights from Multi-Scale Molecular Dynamics Simulations” deals with the excitation energy transfer between Rhodamine molecules under electronic strong coupling conditions. The Authors satisfactorily answered and made corresponding corrections in the revised manuscript. However, some more queries arise from the responses made by the authors.

General Response: We thank the Reviewer for evaluating our manuscript a second time. We are pleased that we could address the previous questions. We have now also addressed the new queries that arose from our previous responses.

Comment 1: The length of the cavity is taken as 163 nm. In such conditions, which mode of the cavity is coupled with the molecular transition? If the fundamental mode of the Fabry-Perot cavity is used, is it matching with the calculated transition energy (4.18 eV) or experimental transition energy (2.5 eV)?

Response 1: The Reviewer wants to know which mode of the cavity was coupled to the molecular transition in our simulations. This is an important point and was indeed not explained with sufficient clarity in the text. As correctly pointed out by the Reviewer, we coupled the *fundamental mode* of the cavity, *i.e.* the mode with the *lowest energy*, with the molecular transition. To compensate for the systematic overestimation of the excitation energy at the level of theory employed in the simulations, we chose the cavity parameters to obtain the minimum energy of that mode at 3.81 eV for $k_z = 0$, which corresponds to a distance between the cavity mirrors of 163 nm. Because with these parameters, the energy of the fundamental mode at normal incidence is 370 meV below the excitation energy of the molecules, its dispersion is red-detuned.

To avoid that readers miss these important details of our model, we have reformulated the text, and emphasized what is the fundamental mode in our cavities.

On page 3: “We computed semi-classical Ehrenfest [62] MD trajectories of 1024 Rhodamine molecules inside a periodic 1D cavity of length $L_z = 50 \mu\text{m}$,

where z indicates the in-plane direction (*i.e.*, parallel to the mirrors). With a distance of $L_x = 163$ nm between the mirrors (cavity width), where x indicates the out-of-plane direction (*i.e.*, perpendicular to the mirrors), the fundamental mode of the cavity has an energy of 3.81 eV at normal incidence (*i.e.*, $k_z = 0$), and hence its dispersion is red-detuned with respect to the molecular excitation energy at 4.18 eV (vertical dashed line in Figure 1b).”

Comment 2: In page 10, the Authors newly added a section about size dependence. Here, it is mentioned that the transport mechanism is not strongly affected by the number of molecules. Then what is the role of collective coupling in these observations?

Response 2: The Reviewer is correct that we mentioned that the transport mechanism is not strongly affected by the number of molecules, but from the comment we realize that our original statement was confusing.

What we tried to convey in the new section about the size dependence was that while the *mechanism* by which an exciton-polariton propagates under the collective strong light-matter coupling in an organic Fabry-Pérot cavity *does not* depend on the number of molecules (N), the *rates* of the processes involved in that mechanism *do* depend on the number of molecules. In particular, because the rate at which population exchanges between the dark and polaritonic states is inversely proportional to N [Tichauer *et al. J. Phys. Chem. Lett.* **13**, 6259 (2022)], we observe that for larger ensembles, with more dark states, the fraction of population residing within the dark state manifold is higher than for smaller ensembles. Such differences affect the (i) propagation velocity (*e.g.*, Figure S14); (ii) the lifetime (*e.g.*, right columns in Figures 4 and S11) and, therefore, (iii) the distance over which the exciton-polaritons are transferred (*e.g.*, compare Figures 5, and S11-S13).

We have reformulated the text in the section on size dependence to differentiate between the effect of N on the mechanism and the effect of N on transport characteristics (see also our response to comment 4, below).

Page 10: “With the exception of the smallest ensemble that lacks dark states, we observe for all other ensemble sizes that the propagation mechanism involves reversible population exchange between the stationary dark state manifold and propagating polaritons. These additional simulations therefore underscore the role of dark states in the propagation process and

suggest that the mechanism does not strongly depend on N . In contrast to the mechanism, however, the rates at which these population exchanges occur, depend on the number of molecules. Indeed, these rates are inversely proportional to N [23,73,77]. Because the number of dark states scales with N , whereas the number of polaritonic states is constant, we observe that for the larger ensembles, the fraction of population residing within the dark state manifold is higher than for the smaller ensembles. Such differences affect (i) the propagation velocity (e.g, Figure S14); (ii) the lifetime (e.g., right columns in Figures 4 and S11); and, therefore, also (iii) the distance over which the exciton-polaritons are transferred (e.g., compare Figures 5, and S11-S13).”

Nevertheless, irrespective of how large N is, collective strong coupling involving multiple molecules and cavity modes, is *essential* to form polaritons in Fabry-Pérot cavities and enhance exciton transport via ballistic propagation of population in the polaritonic states. Such population is either created initially by resonantly pumping a wavepacket of polaritonic states, or transiently, through reversible population transfer from stationary dark states. We re-emphasize this point in our revision (see also our response to comment 4).

On page 10: “Reaching the strong coupling regime to form polaritons with organic molecules in Fabry-Pérot cavities with mode volumes in the order of $V_{\text{cav}} \propto (\lambda_{\text{cav}}/n)^3$ (where λ_{cav} is the wavelength of the cavity mode and n the refractive index), thus requires collective coupling of many molecules (*i.e.*, 10^5 - 10^8 [74-76]).”

Comment 3: The Authors made a rough estimation of the ratio of polaritonic states to Dark states in the response letter and supporting information. The calculated numbers are independent of a molecule under study. It is only about numbers. Is there any role of the strength of strong coupling (Rabi splitting) in determining the ratio mentioned above?

Response 3: The Reviewer wants to understand if the coupling strength, as manifested by the Rabi splitting between the upper and lower polariton branches, plays a role in determining the ratio between the dark and polaritonic states.

Because we determined the ratio between the dark and polaritonic states for a hypothetical situation, in which all molecules are identical and the cavity mirrors are perfect, the polaritonic and dark states are easily identified as

eigenstates with and without contribution from cavity modes, respectively. Furthermore, because in this ideal scenario the bright polaritons and the degenerate dark states form irrespective of the coupling strength (provided it is not zero), their ratio is independent of the Rabi splitting and only depends on N , as $N_{\text{dark}}/N_{\text{pol}} = (N - n_{\text{modes}})/2n_{\text{modes}}$.

In contrast, when there is thermal disorder among the molecules and the molecular excitation energies and transition dipole moments span a distribution, the cavity modes become smeared-out over very many eigenstates of the molecule-cavity system [Houdré *et al.*, *Phys. Rev. A* **53**, 2711–2715 (1996)]. Identifying which states are dark and bright therefore requires a numerical threshold for the total photonic contribution. Such a threshold is arbitrary, but can affect the ratio (see Figure S4 below), and is therefore difficult to choose.

Nevertheless, as shown in panel a of the Figure below, calculating the ratio at constant Rabi splitting (325 meV) for various thresholds, *i.e.*, 0.01, 0.02, and 0.05, with energies and transition dipole moments extracted from the QM/MM equilibrium trajectory at 300 K, suggests that irrespective of the value for the threshold, the ratio between dark and polaritonic states scales approximately linearly with N , but with a slope that depends on the threshold.

Furthermore, the extent to which the cavity mode excitations are smeared-out over the states not only depends on how large the disorder is, but also on the collective coupling strength, as the Reviewer points out [Groenhof *et al.*, *J. Phys. Chem. Lett.* **10**, 5476-5483 (2019), Mony *et al.*, *Adv. Funct. Mater.* **31**, 2010737 (2021)]. Indeed, as shown in panel b of the Figure below, calculating the ratios at different coupling strengths, but for fixed $N = 500$ and a numerical threshold of 0.01, suggest that at the Rabi splitting employed in our simulations (325 meV), the ratio is significantly smaller with disorder than without disorder. However, because we kept the Rabi splitting the same in all our simulations by adjusting the vacuum field strength of the cavity (see response to comment 4 below), and the ratio is proportional to N at constant Rabi splitting (panel a), the dependence on the Rabi splitting does not influence the comparison between different ensembles.

Nevertheless, to avoid complications associated with choosing an arbitrary threshold, we estimated the ratios for systems with different numbers of

molecules, based on the ideal situation without disorder, in which the distinction between polaritonic and dark states is not only unambiguous but also independent of the Rabi splitting and the molecular properties.

Figure S4: In panel **a** the ratio between dark and polaritonic states at constant Rabi splitting (325 meV) is plotted as a function of the number of molecules in a disordered ensemble collectively coupled to a single mode cavity that is resonant with the molecular excitation energy (4.18 eV), for various values of the threshold parameter ε (i.e., $|\alpha^m|^2 > \varepsilon$).⁵ The ratio for a system without disorder is shown in blue. Panel **b** shows the ratio as a function of Rabi splitting for an ensemble with 500 molecules and a threshold of $\varepsilon = 0.01$.

To further motivate our choice for computing the ratio between dark and polaritonic states in systems without disorder, and emphasize that in an disordered system this ratio can be affected by the Rabi splitting, we have expanded the discussion in the Supporting Information. We also included the Figure shown above to illustrate how the ratio depends on N and on Rabi splitting (Figure S4).

On pages 16-17 of SI: “Here, we determined these ratios for an ideal system without disorder, in which all molecules are identical and the polaritonic and dark states can be easily identified. Furthermore, because in this ideal scenario bright polaritons and degenerate dark states form irrespective of the coupling strength (provided it is not zero), their ratio is independent of the Rabi splitting and only depends on N as $N_{\text{dark}}/N_{\text{pol}} = (N - n_{\text{modes}})/2n_{\text{modes}}$.

In contrast, when there is thermal disorder, as in our MD simulations, the cavity modes are smeared-out over many eigenstates.^{21,42,43} In this situation, there is no longer a clear distinction between dark and polaritonic states. While we could in principle apply a numerical criterion and consider a state $|\psi^m\rangle$ polaritonic if the total cavity mode contribution to that state exceeds an arbitrary threshold, ε (i.e., $\sum_k |\alpha_k^m|^2 > \varepsilon$, Equation 14),^{5,21} such an approach would make the ratio dependent on the choice of that threshold.

Nevertheless, as shown Figure S4a, calculating the ratio at constant Rabi splitting (325 eV) for various thresholds, i.e., $\varepsilon = 0.01, 0.02$ and 0.05 , with energies and transition dipole moments extracted from the QM/MM equilibrium trajectory at 300 K, suggests that irrespective of the value for the threshold parameter, the ratio between dark and polaritonic states scales approximately linearly with N , but with a slope that depends on the threshold.

Furthermore, the extent to which the cavity mode excitations are smeared-out over the states not only depends on how large the disorder is, but also on the collective coupling strength, or equivalently, the Rabi splitting.^{21,42} Indeed, as shown in Figure S4b, calculating the ratios at different coupling strengths, but for fixed $N = 500$ and a numerical threshold of $\varepsilon = 0.01$, suggest that at the Rabi splitting in our simulations (325 meV), the ratio is significantly smaller with disorder than without disorder. However, because we adjusted the vacuum field strength to keep the Rabi splitting the same for all systems, and the ratio is proportional to N at constant Rabi splitting (Figure S4a), the dependence on the Rabi splitting does not influence the comparison between the different ensembles. Nevertheless, to avoid complications associated with choosing an arbitrary threshold, we estimated the ratios for systems with different numbers of molecules, based on the ideal situation without disorder, in which the distinction between polaritonic and dark states is not only unambiguous but also independent of the Rabi splitting and the molecular properties.”

Comment 4: Page 10, column 1, the last sentence is unclear.

Response 4: We thank the reviewer for pointing out that this sentence lacks clarity. We agree that more details may be required for the reader to understand the point we are trying to make and have now rephrased the sentence “To keep the Rabi splitting ($\sim 325\text{meV}$) constant, and hence

polariton dispersion the same, we scaled the cavity mode volume with the number of molecules N (see SI for details)” as follows:

On Page 10: “The Rabi splitting depends on the vacuum field strength, \mathbf{E}_y , and the number of molecules, N , via $\hbar\Omega^{\text{Rabi}} = 2 \mu^{\text{TDM}} \cdot \mathbf{E}_y \sqrt{N}$, with μ^{TDM} the molecular transition dipole moment [18], which for organic molecules is on the order of few Debye. Because the vacuum field strength of a cavity is inversely proportional to the square root of the mode volume (Equation 3 in SI), the Rabi splitting scales with the molecular concentration in the mode volume, V_{cav} , of the cavity, *i.e.*, $\hbar\Omega^{\text{Rabi}} \propto \sqrt{N/V_{\text{cav}}}$. Reaching the strong coupling regime to form polaritons with organic molecules in Fabry-Pérot cavities with mode volumes on the order of $V_{\text{cav}} \propto (\lambda_{\text{cav}}/n)^3$ (where λ_{cav} is the wavelength of the cavity mode and n the refractive index), thus requires collective coupling of many molecules (*i.e.*, 10^5 - 10^8 [74-76]). Because the number of molecules that we can include in our simulations is much smaller due to limitations on hard- and software, we investigated how that number affects the propagation by repeating simulations for different N . To keep the Rabi splitting constant, and hence the polariton dispersion the same, we scaled the mode volume with N , *i.e.*, $V_{\text{cav}} = N V_{\text{cav},0}$, where $V_{\text{cav},0}$ is the mode volume required to achieve a Rabi splitting of 325 meV with a single Rhodamine molecule in the cavity.”

REVIEWERS' COMMENTS

Reviewer #1 (Remarks to the Author):

I appreciate the Authors for their detailed response. They satisfactorily answered and significantly modified the texts in the revised version of the manuscript, which will benefit broad readers. Therefore, I recommend the revised version of the manuscript for publication in *Nat. Commun.*